# Early hypogenic Carbonic Acid Speleogenesis in unconfined limestone aquifers by upwelling deep-seated waters with high $CO_2$ concentration: A modelling approach

Franci Gabrovšek[1], Wolfgang Dreybrodt[1,2]

[1] Karst Research Institute ZRC SAZU, Titov trg 2, 6230 Postojna, Slovenia

[2] Faculty of Physics and Electrical Engineering, University of Bremen, Germany

*Correspondence to*: Franci Gabrovšek (gabrovsek@zrc-sazu.si)

**Abstract** Here we present results of digital modelling of a specific setting of hypogenic Carbonic Acid Speleogenesis (CAS). We study an unconfined aquifer where meteoric water seeps through the vadose zone and becomes saturated with respect to calcite when it arrives at the water table. From below deep-seated water with high $p_{CO2}$ and saturated with respect to calcite invades the limestone formation by forced flow. Two flow domains arise that host exclusively water from the meteoric or the deep-seated source. They are separated by a water divide. There by dispersion of flow, a fringe of mixing arises and widening of the fractures is caused by mixing corrosion (MC). The evolution of the cave system is determined by its early state. At sites with high rates of fracture widening regions of higher hydraulic conductivity are created. They attract flow and support one by one mixing with maximal dissolution rates. Therefore, the early evolution is determined by karstification originating close to the input of the upwelling water and at the output at a seepage face. In between these regions, a wide fringe of moderate dissolution is present. In the later stage of evolution, this region is divided by constrictions that originate from statistical variations of fracture aperture widths that favour high dissolution rates and focus flow into this region. This MC-fringe-instability is an intrinsic property of cave evolution and is present in all scenarios studied. We have investigated the influence of defined regions with higher fracture aperture widths. These determine the cave patterns and suppress MC-fringe-instabilities. We have discussed the influence of the ratio of upwelling water flux rates to the rates of meteoric water. This ratio specifies the position of the mixing fringe and consequently that of the cave system. In a further step, we have explored the influence of time dependent meteoric recharge. Furthermore, we have modelled scenarios where waters are undersaturated with respect to calcite. These findings give important insight into mechanisms of carbonic acid speleogenesis (CAS) in a special setting of unconfined aquifers. They also have implications to the understanding of corresponding sulphuric acid speleogenesis (SAS).

# 1 Introduction

Hydrochemical digital models of speleogenesis are powerful tools to understand the physical and chemical processes that determine the evolution of caves. Initial models of the evolution of 1D fractures (Dreybrodt, 1990;Palmer, 1991;Dreybrodt, 1996) were soon extended to different manifestations of fracture networks within 2D and 3D domains (Groves and Howard, 1994;Siemers and Dreybrodt, 1998;Kaufmann et al., 2010;Kaufmann, 2016;Li et al., 2020). An example variety of hypothetical hydrological, structural and geochemical settings were envisioned in order to understand the basic speleogenetic mechanisms (Birk et al., 2003;Kaufmann, 2003;Dreybrodt et al., 2005;Gabrovsek and Dreybrodt, 2010). The models have also been used for a process-based interpretation of real situations (Kaufmann and Romanov, 2019). The development of the models has been guided by theoretical insights based on small scale detailed studies of dissolution front propagation and fingering (Hanna and Rajaram, 1998;Cheung and Rajaram, 2002;Szymczak and Ladd, 2009;Dreybrodt and Gabrovšek, 2019). Most of the models simulate epigene speleogenesis governed by the aggressive solution from the surface.

In the last two decades hypogenic caves have attracted broad interest in the karst community highlighting in the recent book "Hypogene Karst regions and caves of the world" (Klimchouk et al., 2017). The book gives a wide overview of hypogenic caves worldwide.

There are two concepts of the evolution of hypogenic caves: **1)** Klimchouk (2007;2016) suggests a *hydrological approach* stating: "*the formation of solution-enlarged permeability structures (void-conduit systems) is caused by fluids that recharge the cavernous zone from below, driven by hydrostatic pressure or other sources of energy, independent of direct recharge from the overlying or immediately adjacent surface*". This definition applies to large gypsum cave systems of the Western Ukraine. These caves motivated a series of studies modelling artesian hypogenic speleogenesis (Birk et al., 2003;Birk et al., 2005;Rehrl et al., 2008;Kaufmann et al., 2014;Li et al., 2020). Thermal water saturated with respect to calcite that rise from below gain renewed aggressiveness when cooling and may create caves. Such scenarios have been modelled by several authors (Andre and Rajaram, 2005;Rajaram et al., 2009;Chaudhuri et al., 2009;Chaudhuri et al., 2013;Gong et al., 2019) and also fit to speleogenetic concept as defined by Klimchouk.

**2)** Palmer (2000;2007) suggests a *geochemical view*, which defines hypogenic caves as "*those formed by water in which the aggressiveness has been produced at depth beneath the surface, independent of surface or soil $CO_2$ or other near-surface acid sources.*" This definition results from observations in Carlsberg Caverns, New Mexico, USA, where sulphuric acid is dominating dissolution of limestone and from caves in the Black Hills, South Dakota, USA, that are created by carbonic acid speleogenesis (Palmer, 2017).

Dissolution by sulphuric acid (Palmer, 2013) can occur where $H_2S$ bearing deep- seated waters rise from below and mix with oxygenated groundwater. There, $H_2S$ is oxidized by bacterial aid to sulphuric acid that dissolves limestone, releasing $CO_2$ for further dissolution of carbonate rock. This speleogenetic process, termed sulphuric acid speleogenesis, (SAS) has created large caves, e.g. Carlsbad Caverns in the Guadalupe reef complex in the United States and the Frasassi cave system in Central Italy.

Carbonic acid operates also as a hypogenic agent that is produced at depth, e. g. by thermal decomposition of carbonate rocks. This way $CO_2$ containing water with high $p_{CO2}$ up to 1 atm can intrude into upper aquifers, especially in areas of young volcanism, point-wise or disperse, depending on geologic/structural conditions (Jeong et al., 2005;Palmer, 2007;Klimchouk, 2013;Audra and Palmer, 2015). $CO_2$ may also stem from oxidation of deep-seated organic compounds, as they are abundant near hydrocarbon fields (Klimchouk, 2019).

It should be mentioned that modelling of dissolution of calcium carbonate in mixing zones in karst systems has been studied in general alternative models by other groups successfully. This approach decouples the solute transport and chemical reaction in the mixing fringes (De Simoni et al., 2007). Mixing of freshwater and seawater in coastal aquifers and the resulting evolution of porosity has also been modelled (Romanov and Dreybrodt, 2006;Laabidi and Bouhlila, 2015). But, to our knowledge such an approach has not been used to model problems of SAS or CAS.

In this work, we focus to a specific hypothetic case of hypogenic carbonic acid speleogenesis (CAS). We study an unconfined aquifer where meteoric water seeps through the vadose zone and becomes saturated with respect to calcite when it arrives at the water table. From below, deep-seated water with high pCO2 and saturated with respect to calcite invades the limestone formation by forced flow.  Mixing corrosion acts at the regions where these waters mix creating cave conduits. From the results, we also discuss analogies with SAS and the problems that arise in modelling of SAS.

**2. The model**

**2.1. Modelling domain of an unconfined aquifer with mixing of surface and deep-seated waters**

As the first step, we discuss the modelling domains and settings that determine hypogenic karstification. Figure 1 shows the settings of an unconfined aquifer. With respect to surface water leaking to the aquifer this setting is similar to the geology of Wind Cave in the Black Hills, South Dakota, USA (Palmer, 2017). In the abstract, he states: "*Cave enlargement depended*
*mainly on diffuse recharge through overlying sandstone, mixing with lateral inflow through carbonate outcrops.*"
This stresses the geological relevance of our model.
A net of fracture conjunctions as depicted in a) each 2 m long and 1 m wide are connected to form a rectangular array that presents a vertical 2D section of an unconfined aquifer of a selected depth and length. To each fracture, individually a selected aperture width is assigned.  This way a net with a lognormal distribution of aperture widths, as shown in b) is created. The
following boundary conditions are applied: At the top of the domain, we impose a constant input, q, of water to each fracture to simulate the recharge by meteoric water. To each input fracture, a defined Ca-concentration and $p_{CO2}$ of the inflowing water is assigned.
At the left hand side, we impose no flow condition to model a water divide. The right hand side presents the outflow of water where each fracture is open to the atmosphere in the upper part (seepage face. The lower region is at no flow condition.
Therefore, a water table arises that divides the aquifer into a vadose (light grey) and a phreatic zone (dark grey). The bottom

boundary of the modelling domain is regarded impermeable (no flow condition, red) with the exception of a defined region (blue) where waters from the depth with defined chemical composition invade the aquifer with constant flow, Q, into each fracture. For each fracture where water enters into the aquifer, Ca-concentration and $p_{CO2}$ are selected. In a pure $CaCO_3$-$CO_2$-$H_2O$ solution, these two parameters define all its chemical properties.


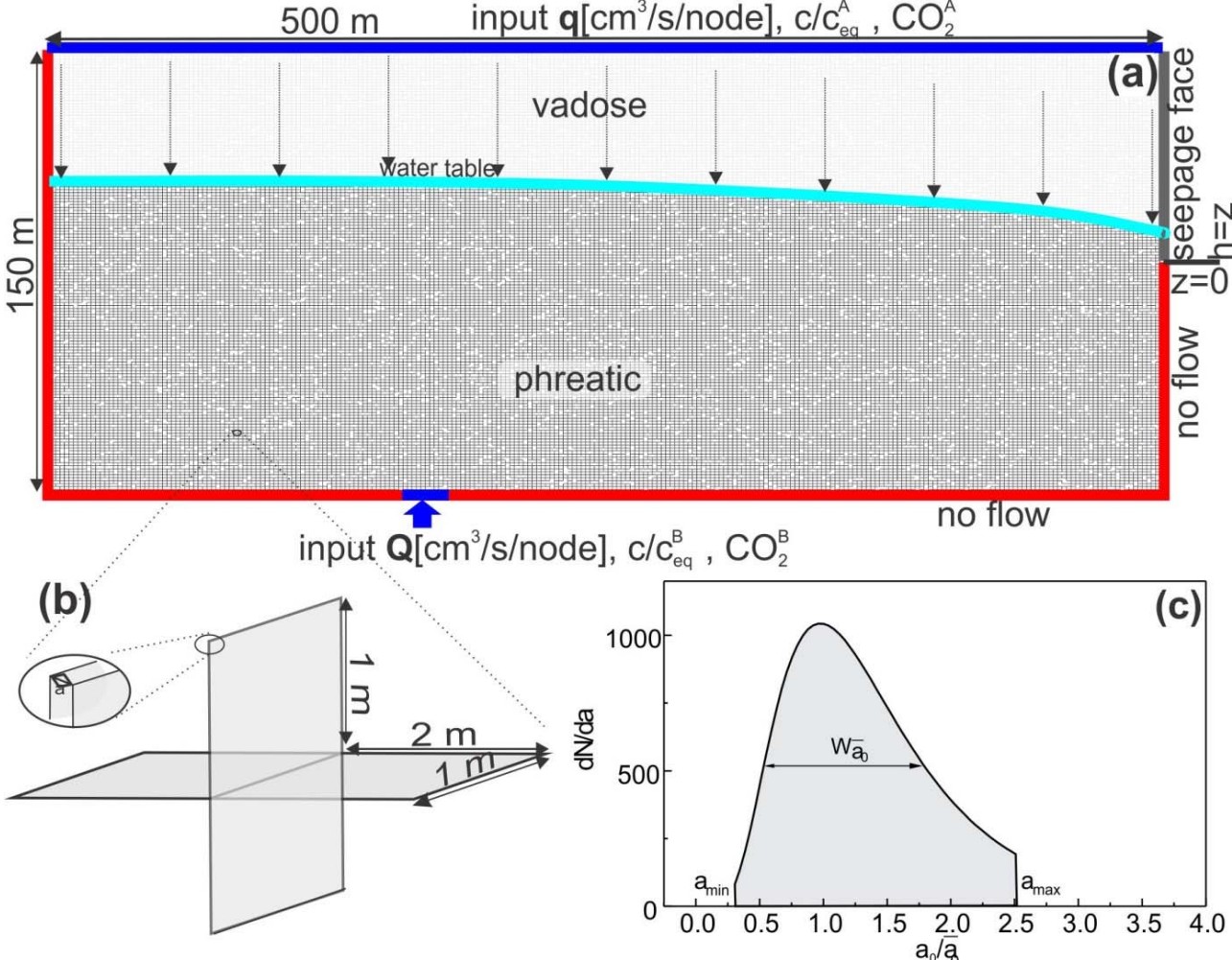

**Figure 1: a) 2 dimensional fracture network presenting a vertical cross-section through unconfined aquifer. The length of the domain is 500 m, the depth is 150 m. Fracture spacing is 2 × 1 m² as depicted in b). Input q, cm³/(s node) at the top is evenly distributed meteoric water percolating to the water table. When it arrives there, it has become saturated with respect to calcite and $p_{CO2}$. At the bottom of the domain deep-seated upwelling water enters with prescribed flow Q, cm³/(s node) by forced flow. Due to dissolution of limestone on its way from below this water is saturated with respect to calcite and high $p_{CO2}$. The water flows to a seepage face located above height z = 0. Red border denotes no flow conditions, blue border denotes input of prescribed flow. Each fracture has an initial aperture width $a$, which is selected from a truncated log-normal distribution, shown in (c).**

There is an important general property of flow under such boundary conditions. Whenever Darcy flow originates from different recharging sources, in our case, meteoric water to the water table and upwelling deep water from below, each source creates a flow domain that hosts only water originating from the corresponding recharge source. Flow domains of different origin of waters are separated by a water divide. The waters of differing origin do not mix. Only at the water divide mixing is possible caused by dispersion of flow into the fractures close to the water divide.

Fig. 2 presents an illustration. Water entering from below with equilibrium concentration $c_{eq}^{B}$ is coloured in red whereas recharge from the top with equilibrium concentration $c_{eq}^{a}$ is coloured in blue. Clearly two flow domains are visible. Water entering from below occupies the lower red region expanding from its input to the outflow at the seepage face. Water from the top (red) that recharges at the seepage face floats on the flow domain below. At the border between the two domains a fringe with rainbow colours is visible that indicate the mixing zone. The colours symbolize the equilibrium concentrations of the mixed solutions and therefore also the mixing ratio as depicted by the colour code. Red is $V_A/V_B = 0/1$, green, $V_A/V_B = 1/1$, dark blue, $V_A/V_B = 1/0$.

The position of this mixing zone depends on the input ratio $q/Q$. With increasing ratio, the mixing zone shifts downwards. Therefore, at a constant inflow from below but varying recharge of meteoric water as it occurs likely during the evolution of caves, the mixing zone fluctuates up and down. When the water entering into the limestone aquifer is undersaturated with respect to calcite we find two regions of limestone dissolution. Each of them extends from the input into the corresponding flow domains but stays limited there. There is also dissolution in the mixing zone where waters that have become saturated mix. However, even in the case when the input waters are saturated with respect to calcite mixing corrosion (MC) is active in the mixing fringe.

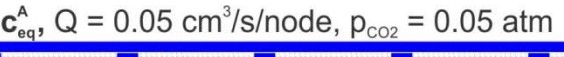

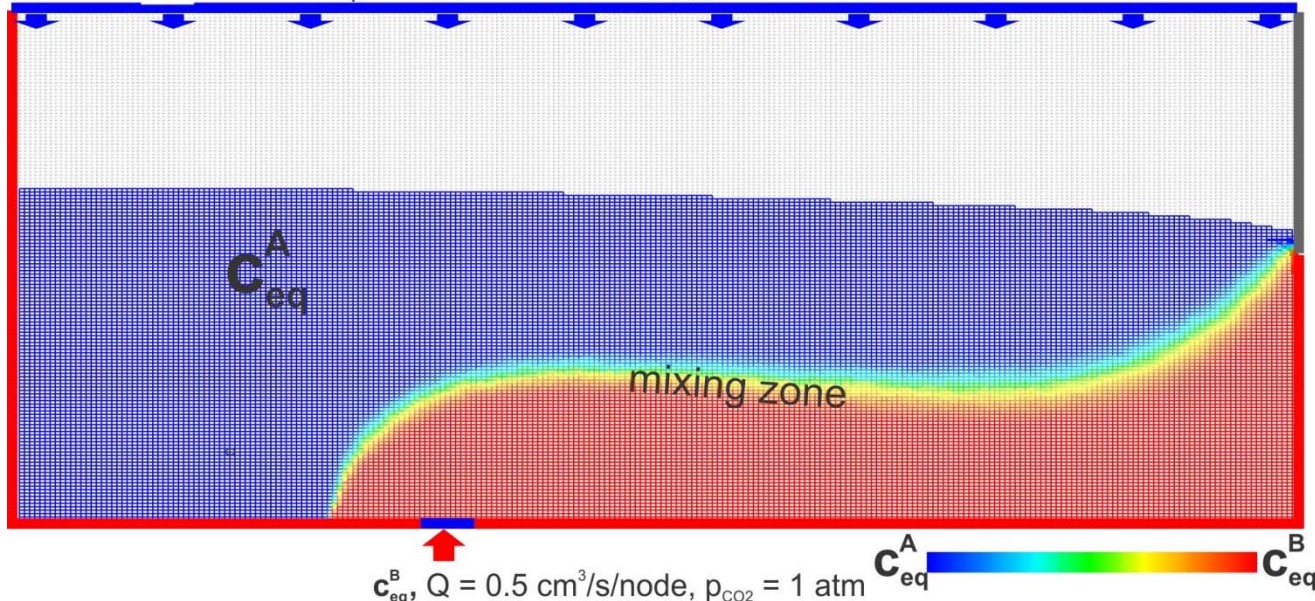

**Figure 2: Visualization of the mixing zone. Two types of waters, one with calcium equilibrium concentration $c_{eq}^{B}$, and the other with equilibrium concentration, $c_{eq}^{b}$ invade the aquifer. Flow domains, red and blue are separated by a water divide. There dispersive mixing takes place and the equilibrium concentration of the mixed solution is between the values $c_{eq}^{B}$ and $c_{eq}^{A}$ (see Fig. 3). This is depicted by a colour code with red for the highest concentration $c_{eq}^{B}$ decreasing down through the optical spectrum to dark blue for concentration $c_{eq}^{A}$. This way the colours indicate the degree of mixing. The green region hosts one by one mixing.**

## 2.2. The digital model

All information concerning the numerical scheme and the model is reported in previous literature: Details of the digital model are given by Dreybrodt et al. (2005), Gabrovšek and Dreybrodt (2010) and Dreybrodt and Gabrovšek (2019).The extension to unconfined aquifers is described in detail in Gabrovšek and Dreybrodt (2001, 2004). The 1-D transport-dissolution model is reported in detail by Dreybrodt (1996). Therefore, we give only brief essentials here.

The simulation in time proceeds through a series of steady states because the time scale of fracture widening by dissolution is much larger than that of transient flow and transport. At each time-step, stationary solutions of flow, transport, and dissolution rates are used to calculate the change of fracture geometry within the time-step. We first calculate the hydrodynamics, based on mass conservation at the fracture intersections, the head loss relation for laminar flow along the fractures and the given boundary conditions. This yields a set of linear equations for the junction heads, which is solved by preconditioned Conjugate

Gradient method. To account for the free upper boundary of the unconfined aquifer, the calculation of heads is wrapped into an iterative procedure seeking that the head of junctions at the water table is equal to their elevation above the base level (Gabrovšek and Dreybrodt, 2001;Gabrovsek et al., 2004).

We specify the calcium concentration $c_{in}$ and $p_{CO2}$ of the inflow solution at all input points. From these the equilibrium
concentration, $c_{eq}$, with respect to calcite is calculated for closed system conditions with respect to $CO_2$. This way MC is included automatically. We calculate the dissolution rates in the fracture draining the input points by the rate law

$$F(c) = k_1 \left(1 - c/c_{eq}\right)$$ with $k_1$=4×10$^{-11}$ mol cm$^{-2}$ s$^{-1}$ for $c < 0.9c_{eq}$ (Dreybrodt et al., 2005;Buhmann and Dreybrodt, 1985). For

$c > 0.9c_{eq}$ a nonlinear rate law $$F(c) = k_4 \left(1 - c/c_{eq}\right)^4$$ with $k_4 = 4×10^{-8}$ mol cm$^{-2}$ s$^{-1}$ is applied applied (Svensson and

Dreybrodt, 1992;Eisenlohr et al., 1999). Next, we use the 1-D transport-dissolution model (Dreybrodt, 1996) to calculate the
calcium concentration profile along all fractures, including the concentration of the solution leaving them. By following the order of decreasing heads, we select all nodes, where the concentrations of the inflowing solutions are known. We assume complete mixing of these solutions in the junctions before they enter into the conduits transporting the flow away. We repeat this procedure until the input concentrations for all fractures are determined. From this, the new profiles of the fractures after a time step $\Delta t$ are obtained by applying the rate laws of dissolution. Then, the new flow rates are calculated and we repeat the
entire procedure to obtain the temporal evolution of the net until some defined condition is met.

## 3. Mixing corrosion

In our Standard Scenario, we assume that the meteoric water after percolating through the vadose zone has become saturated with respect to calcite when it reaches the water table. The upwelling water on its way from below also has reached saturation. Therefore widening of fractures by dissolution of limestone is possible only in the mixing zone by mixing corrosion (MC).
For better understanding, mixing corrosion in the most general case is explained as follows.

Figure 3 depicts a plot of the Ca- equilibrium concentration in its relation to the $p_{CO2}$ in the solution. Whether a $H_2O$-$CO_2$-$CaCO_3$ solution is dissolving calcite can be judged from this equilibrium line. The equilibrium line (black) is given by thermodynamics as (Dreybrodt, 1988) $c_{eq} = A(T) p_{CO_2}^{1/3}$ where $T$ is the temperature of the solution and $p_{CO_2}$ is the partial pressure of $CO_2$ in equilibrium with the solution. $A(T)$ is a constant depending on temperature only. A(T) = 11.3, 9.45, and
7.93 mmol L$^{-1}$(atm)$^{-1/3}$ at 10, 20, and 30°C respectively.

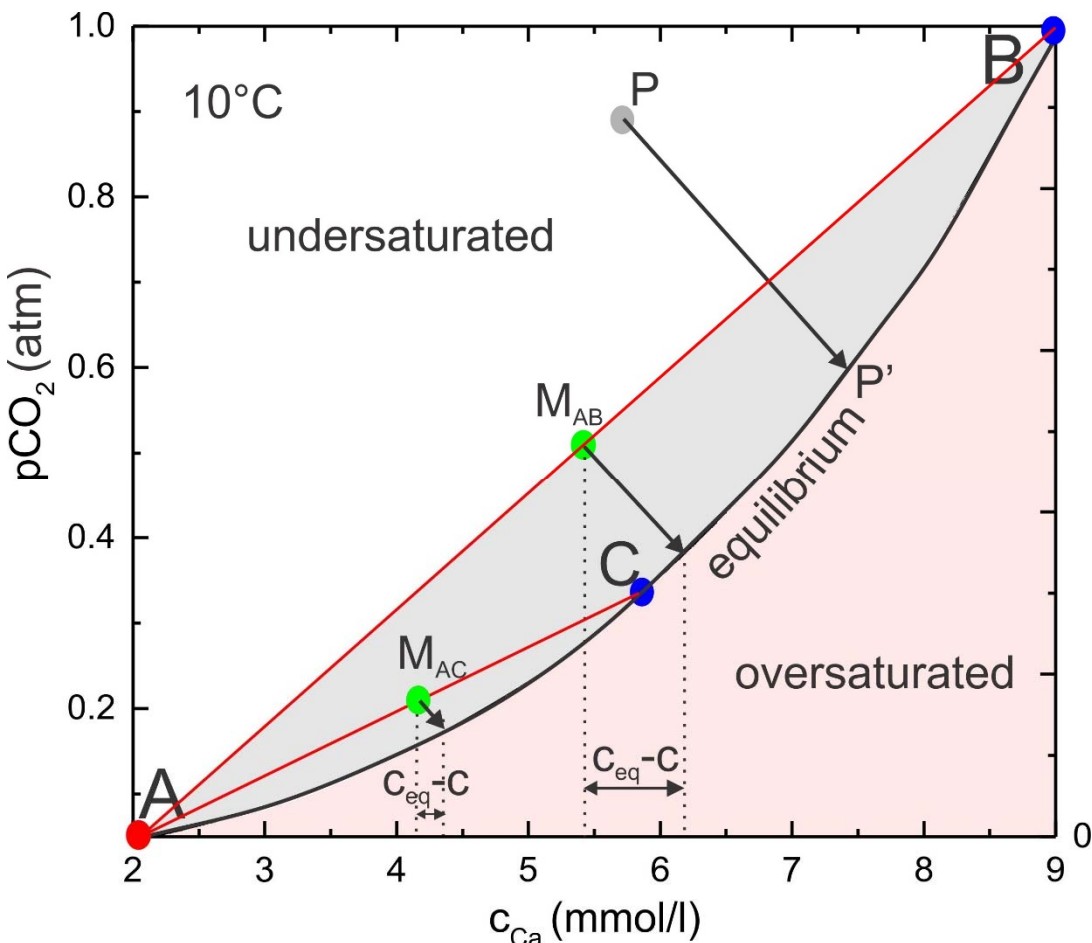

**Figure 3: Processes in mixing corrosion (MC), see text.**

The processes in MC are illustrated for $T = 10°C$, in Fig. 3. The chemical composition of each solution in a $H_2O\text{-}CO_2\text{-}CaCO_3$ system is defined by the ($c_{Ca}$ , $p_{CO_2}$) coordinates. Each solution with chemical composition as given by point P above the equilibrium line (black) is undersaturated with respect to calcite whereas each point below the black equilibrium line is oversaturated and calcite can be precipitated from such a solution. Dissolution in fractures proceeds under closed system conditions where for each $CaCO_3$-unit dissolved one molecule of $CO_2$ is removed from the solution. The straight black arrows

depict the reaction pathway under closed system conditions as used in all our scenarios. Equilibrium with respect to calcite is obtained, for example, when the line from P reaches point P'.

   Classical MC, as a special case is defined when both, the mixing solutions A and B or C are saturated with respect to calcite. Due to the curvature of the equilibrium line, the mixed solutions $M_{AB}$ and $M_{AC}$ on the corresponding mixing lines (red) are undersaturated and can dissolve calcite. The amount of $CaCO_3$ dissolved can be read easily from the dotted arrow between the

vertical dotted lines for the mixed solutions. To calculate $c_{eq}$, evolving from any point P, $M_{AB}$, or $M_{AC}$ one has to find the intersection of the reaction path lines with the equilibrium curve. This yields a cubic equation for $c_{eq}$.

## 4. Standard Scenario: Pure Mixing Corrosion

Figure 4 illustrates karstification of our Standard Scenario. The colour code depicts the rate of fracture widening in cm/Ky.
Red is the maximal rate of 10 cm/ky, black is zero. The aperture widths of the fractures are shown by the bar code. The vadose zone is shown in light gray and the phreatic zone in black (no dissolution) or in colour (dissolution active).

Dissolution is active by mixing corrosion in the fringe where meteoric water and water from below mix (see also Figs. 2 and 3). At the beginning (Fig. 4a), a fringe of moderate dissolution rates increases the hydraulic conductivity in this region.

After 500 years (Fig. 4b) high dissolution rates are seen (orange) close to the input region of hypogenic
water. Below that region of enhanced dissolution, one finds black widened fractures where dissolution has stopped. Evidently, the mixing zone due to the change of hydraulic properties has moved upwards. In the middle section, the region of mixing has become wider. Close to the exit of water dissolution is restricted to a narrow band with high dissolution rates.

After 5000 years (Fig. 4c) the region of dissolution has moved further upwards leaving a region of widened fractures below (black). A main channel has formed to which dissolution is restricted. It widens uniformly and stays stable as can be seen at
30000 years of evolution (Fig. 4d).

This is also illustrated in Fig. 5 that shows rates of fracture widening (green) in cm/year and the aperture widths (red) of the horizontal fractures along the vertical transects $p_1$, $p_2$, and $p_3$ as depicted in Fig. 4a. Although there is some variation in the fracture widening its region is restricted to a depth of 100 to 113 m where the rates stay almost constant in time with maximum rates of about several cm/ky. Therefore, the aperture widths increase linearly in time to a width of about 50 cm after 30000
years.

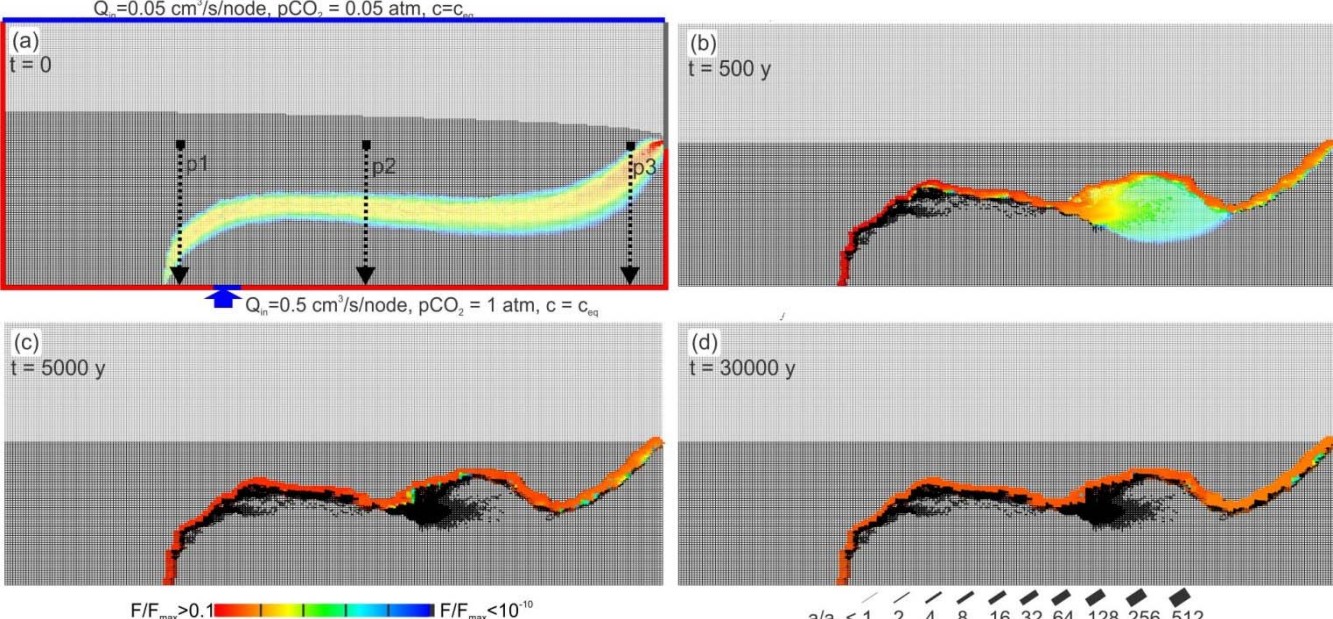

**Figure 4: Temporal evolution of the Standard Scenario with MC exclusively. The colour code depicts rate, F, of fracture widening divided by the maximal widening $F_{max}$ = 10 cm/ky in the net. Aperture widths, a, of fractures are shown by a bar code in units of $a_0$, the initial average aperture width. After 5000 years the pattern is stable in time and all fractures widen with rates almost constant in time. $P_1$, $P_2$, $P_3$ are the positions of the transects of fracture aperture widths and rates of fracture widening shown in Fig. 5.**

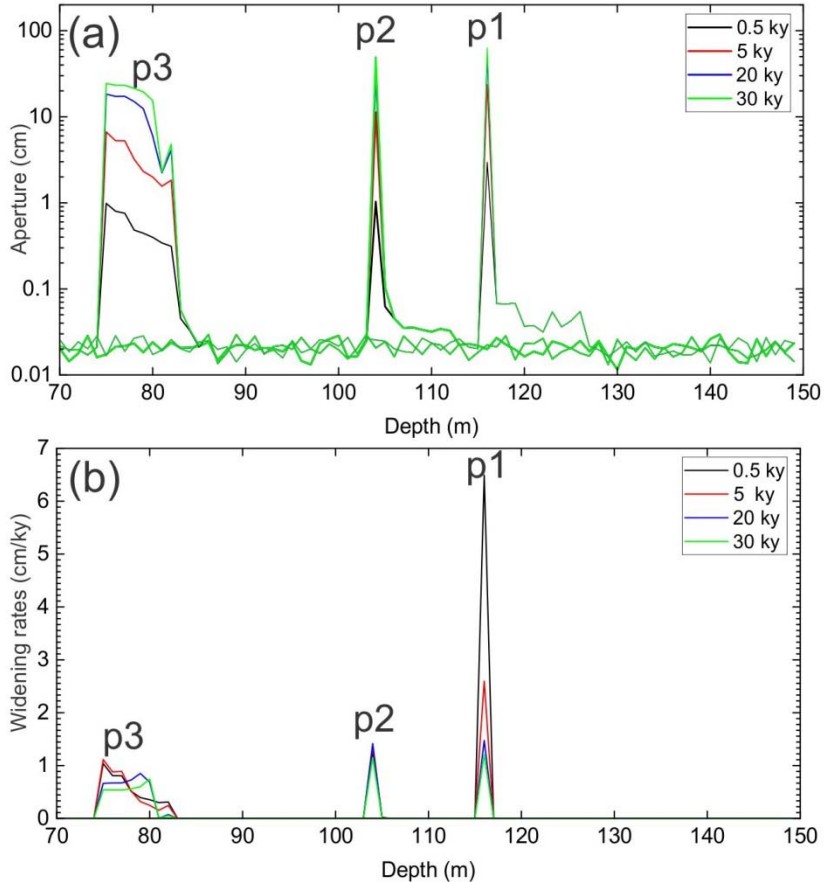


**Figure 5: Transects of aperture widths and rates of widening of the horizontal fractures along the positions p₁, p₂, and p₃ in Fig. 4a**.

To explore the influence of the $p_{CO2}$ in the upwelling water we have performed a simulation with $p_{CO2} = 0.2$ atm and everything
else as in the Standard (not shown). The result is very similar to the Standard. But due to the smaller $(c-c_{eq}) = 0.1$ mmol/l at
0.2 atm (see Fig. 3) dissolution rates are reduced by almost one order of magnitude compared to 0.8 mmol/l at 1 atm (see Fig.
3). Consequently, the evolution to similar conduit (fracture) dimensions is delayed accordingly.

**5. Influence of the ratio q/Q**

As already stated, the position of the mixing fringe depends on the input ratio $q/Q$. Therefore, we have studied the evolution
with three different recharge rates of rain water corresponding to 400, 800, and 1600 mm/year and constant input of hypogenic
water from below. Figs. 6a, b, and c exhibit the distribution of dissolution rates and fracture aperture widths after 44000 years.
In all cases, one finds a fringe of karstification that is related to the location of the mixing zone. With increasing recharge by

meteoric water this fringe is shifted downwards. Figure 7 depicts the profiles of the aperture width of the horizontal fractures

close to the input of hypogenic water and close to the output at the spring. The profiles of aperture widths are very similar for all three cases, because the mixing of the waters does not depend on the position of the mixing zone.

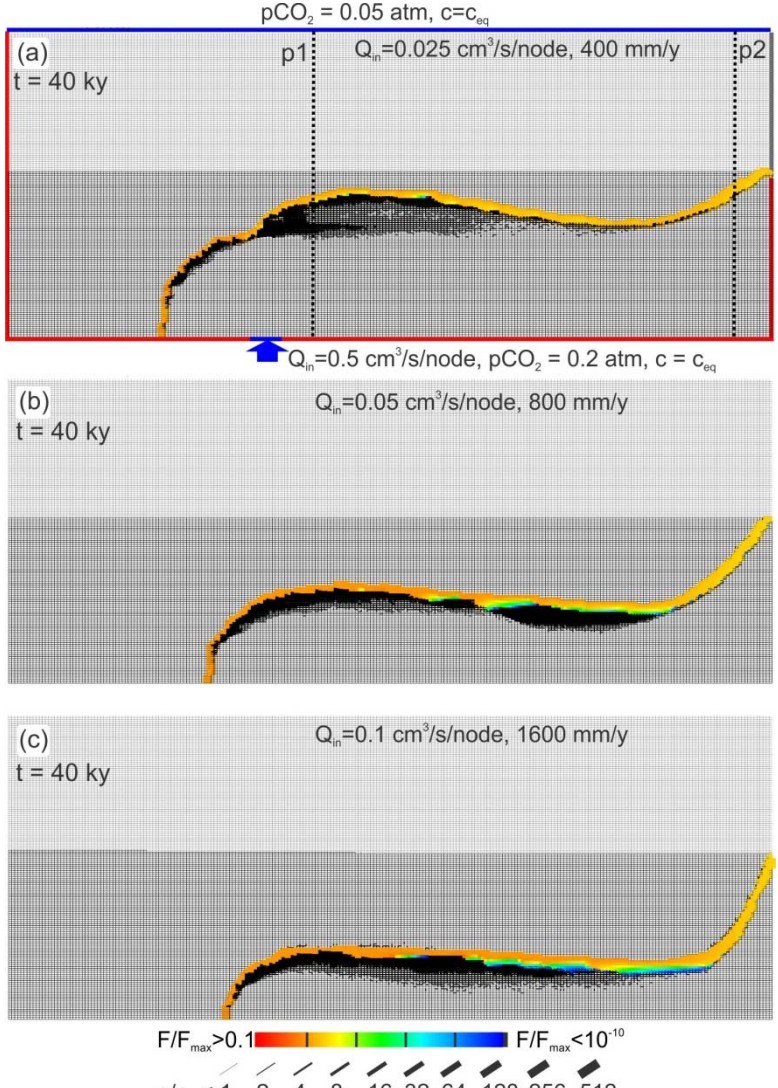

Figure 6: Cave pattern after 40000 years for various recharge rates Q, a) 400 mm/year, b) 800 mm/year, c) 1600 mm/yea

and constant recharge from below. With increasing recharge, the flow domain of meteoric water increases and the fringe of mixing is shifted downwards. To show that even low $p_{CO2}$ in the upwelling water creates substantial karstification we have used $p_{CO2} = 0.2$ atm. Everything else as in Standard Scenario.

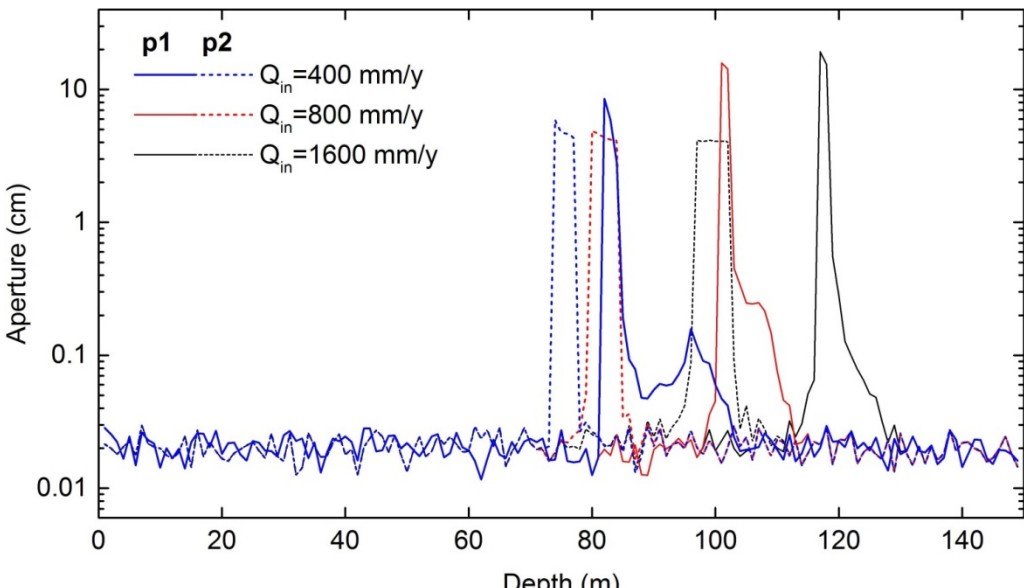

**Figure 7: Vertical profiles of horizontal fracture aperture widths along the positions p1 and p2 shown in Fig. 6. after 40 ky of evolution.**

## 6. Standard Scenario. Meteoric recharge changes in time

In all cases treated so far we have assumed that meteoric recharge is constant in time. This assumption is not realistic. In nature, meteoric recharge does fluctuate and consequently the location of the mixing fringe will change its position accordingly. Therefore karstification may not be longer limited to the initial narrow fringe but could also be active between the fringe positions of maximal, $Q_{max}$, and minimal, $Q_{min}$, meteoric recharge.

To reveal the evolution under such conditions we studied the following scenarios. In the first, meteoric recharge is low at 400mm/year during the first 10,000 years of evolution. Then recharge is switched to a high value of 1,600 mm/year. In the second scenario we start with the high recharge and switch to the low one after 10,000 years. The results are shown in Fig. 8. The panels on the left hand side illustrate the temporal evolution when the initial input is low. In the beginning, the mixing zone is close to the water table due to the small input of rainwater. Dissolution and correspondingly widening of the fractures is restricted to this region, which gains higher hydraulic conductivity than its neighboring regions. This focuses flow of both, the flow domain from the input below and the flow domain of the rain water into the region of increased hydraulic conductivity during the first 500 years. A channel develops reaching the dropping water table and then after looping down approaches the spring. This channel attracts all flow and even when the model switches to higher input of rainwater after 10,000 years the further evolution proceeds along this channel as depicted in the lowest panel after 15,000 years.

The right hand side panels show the temporal evolution when the initial input of meteoric water is high at 1600 mm/year and then after 10,000 years switches to the low value of 400 mm/year. Due to the high input the mixing zone is forced deeper into

the aquifer. In analogy to the left hand side scenario in this region high conductivity is created that further determines the evolution of a channel

deeper in the aquifer that does not change its position even when the input of rain later is reduced drastically after 10,000 years. In summary: The evolution of caves is highly determined by its initial phase of evolution and restricted to the regions where widening of fractures occurs initially. This is due to a positive feedback loop that exists initially. Widening of fractures attracts flow increasing dissolution rates that cause widening of fractures and so on. However, this feedback exists only in the beginning when the rates are low and breaks down later when the dissolution rates remain finite.

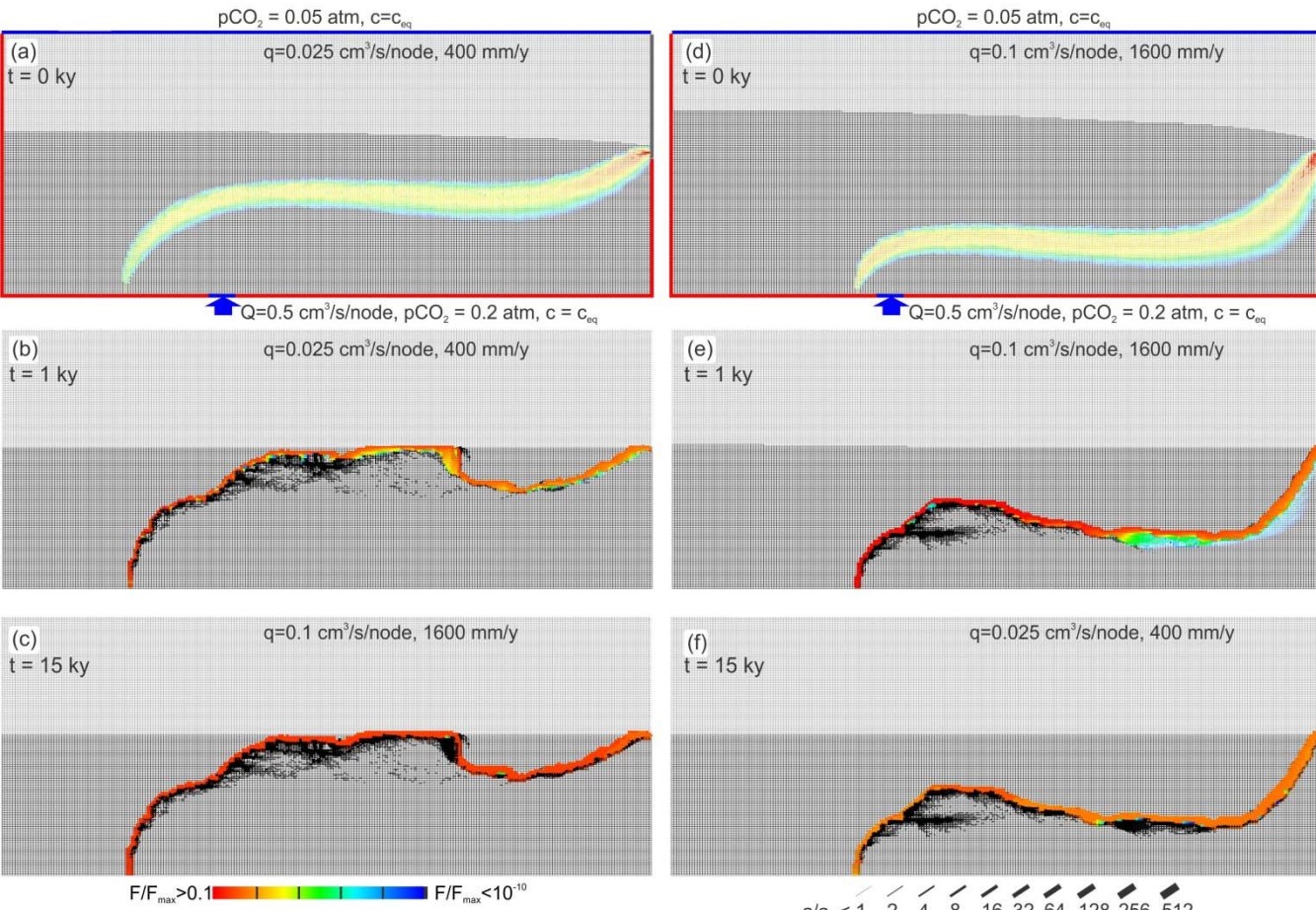


**Figure 8. Temporal evolutions when meteoric recharge changes in time. Left hand side panels: Initial input is 400 mm/year and stays constant for 10000 years, and then it switches to 1600 mm/year constant in time. Right hand side panels: Initial input is 1600 mm/year and stays constant for 10000 years, and then it switches to 400 mm/year constant in time.**


To illustrate this mechanism we have explored the following scenario. We have inserted a region where all fractures have initial apertures of $a_0 = 0.04$ cm as indicated in Fig. 9a. Fig. 9b depicts the evolution after 1200 years. The right hand side column shows the evolution of the corresponding scenario without the widened region (Standard Scenario, Fig. 4). It is clearly seen that the zone of widened fractures attracts flow. The mixing zone is pushed upwards and the conduits of the cave system are located in this zone of wide fractures. It should be stressed at that point that regions with doubling of the initial aperture widths are sufficient to change the patterns of the cave systems drastically. Average initial widening of aperture widths is about $10^{-4}$ cm/year in all scenarios discussed in this work. Accordingly, only the very early state (200 years) of cave evolution is sufficient to create regions of doubled aperture widths that determine the future evolution of the cave system.

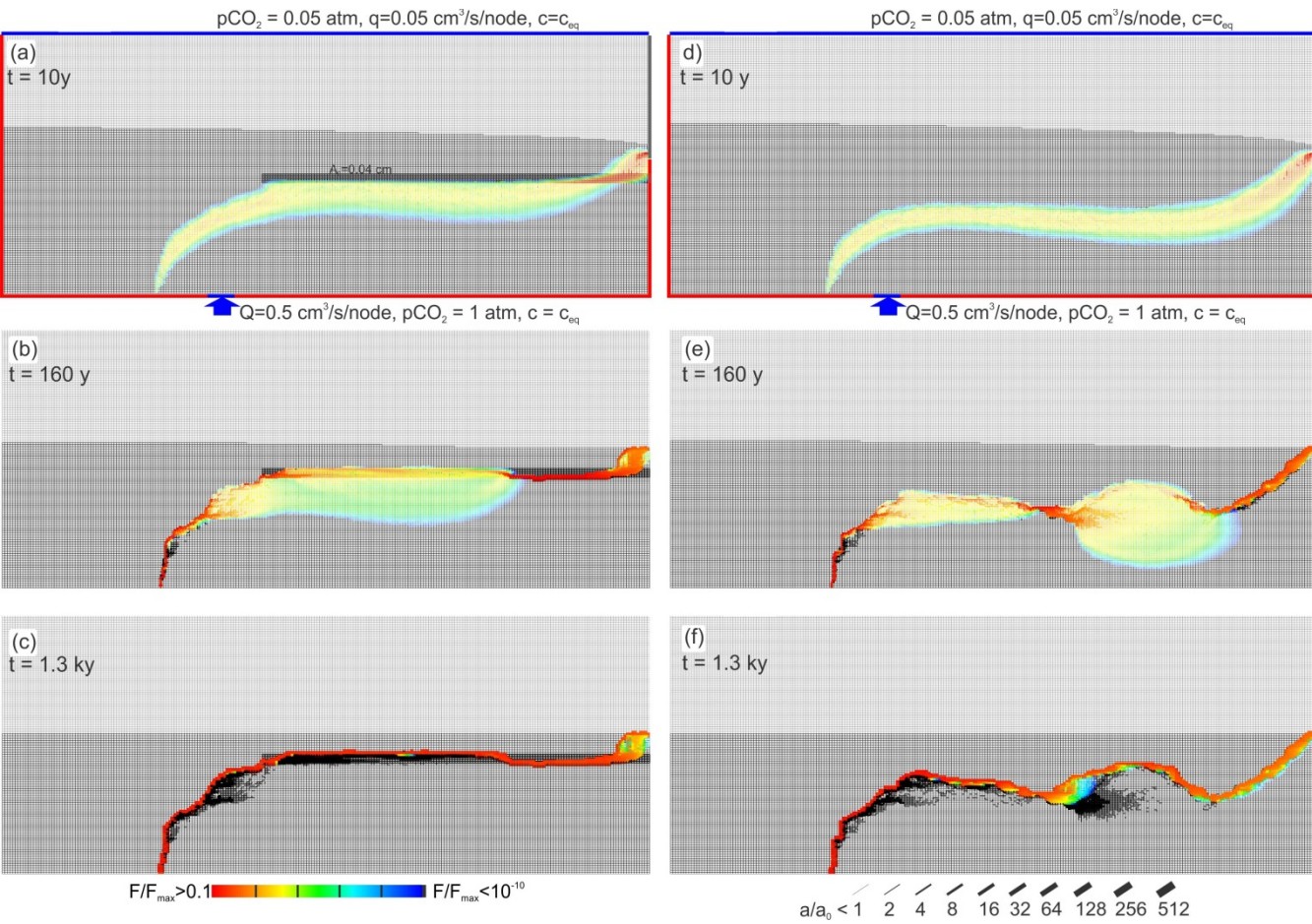


**Figure 9: Evolution of the Standard Scenario with region fractures with larger initial aperture ($a_0 = 0.04$ by black region, a-c) compared to the evolution of the Standard Scenario on the right hand side (d-f). The region of enlarged fractures dominates the evolution.**

## 7. Standard Scenario with input of aggressive water

So far we have assumed that the upwelling water is saturated with respect to calcite. If, however, the deep water on its way upwards does not pass any limestone formation it will be undersaturated with respect to calcite. In the extreme case, its calcium concentration is zero. We have modeled this extreme case for the conditions $q = 0.05$ cm-3/node corresponding to 800 mm/year, $Q = 5$ cm3/node $c_{in}/c_{eq} = 1$ and $p_{CO2} = 0.05$ atm at top $c_{in}/c_{eq} = 0$ and $p_{CO2} = 1$ atm. at the bottom. The temporal evolution is shown in Fig. 10.

**Figure 10: Standard Scenario with upwelling water undersaturated with respect to calcite, cin = 0, and input raised to 5 cm3/s. The red region indicates high rates of fracture widening.**

The flow domain of the water from below extends close to the water table. There MC is active as indicated by red fringe. Due to the high hydraulic conductivity created by the aggressive water from below the water divide drops to the highly conductive region where further dissolution remains active.

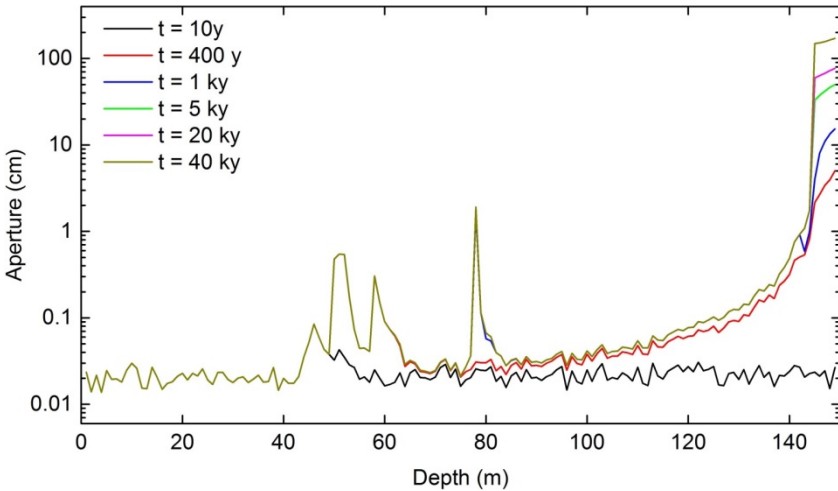

**Figure 11: Aperture widths of horizontal fractures along a vertical profile at position marked by dashed line (p) in Fig. 10a. Note that in the late evolution widening of fractures is restricted to the fractures at base level.**

In the initial state (Fig. 10a) two regions of dissolution are seen. The upwelling water is highly aggressive and creates a region (red) of high dissolution rates in the input region. More distant from the input this water approaches saturation as one can see from the change of colour from red to blue. Due to the high input rate of the upwelling water, the water table is initially higher than in the standard scenario.

At the water divide between the flow domains the upwelling water is mixed with the meteoric water from the surface and gains renewed aggressivity by MC as can be seen from the red fringe in the mixing zone. The mixing zone is initially above the base level. Due to increasing hydraulic conductivity the mixing zone drops as karstification proceeds, leaving behind the widened inactive regions in the vadose and phreatic zone (Fig. 10b). After 1000 years (Fig. 10c) the region of dissolution has dropped to the base level. The mixing zone is limited to the outflow region. Dissolution is active only in the region at the bottom of the aquifer where inflowing water from below is guided due to the high conductivity that has been created there. At the end of its horizontal part the water rises still maintaining its dissolution power. The flow domain of the water from below is restricted to the region of highly widened fractures. In the black region of the aquifer dissolution has stopped leaving a complex pattern of widened fractures. This structure of the aquifer is stable in time as can be seen from the panel at 40000 years (Fig. 10d). In the red region fractures widen continuously creating complex caves. The aperture widths of the horizontal fractures along vertical transects as indicated in Fig. 10 a are shown in Fig.11.

**Figure 12: Standard Scenario, but upwelling water is undersaturated with respect to calcite, $c_{in}$ = 0. The red region indicates high rates of fracture widening. The highly aggressive water is channeled to the mixing zone at the water divide.**

In order to find the influence of flow rate, Q, of the upwelling water we have reduced Q from 5 cm- 3/(node and s) to 0.05 cm/(node and s) and have left everything else as in Fig. 10. The result is shown in Fig. 12. Because of the reduced input, Q, the region of high dissolution rates (red) is much smaller and the mixing zone at a lower position compared to Fig. 10. After 200 years (Fig. 12b) dissolution is active in both, the plume of upwelling water and in the mixing zone. At 800 years (Fig. 12c) flow is mainly along the channel that has evolved along the mixing zone and only there do the fractures widen. This pattern remains stable as can be visualized after 10,000 years (Fig. 12d)

To complete our understanding of dissolution patterns we have addressed the question: what happens when both, the meteoric surface water and the upwelling water from below are undersaturated with respect to calcite. To study this systematically we compare the results of three scenarios:

**Scenario A)** The meteoric water that enters the water table is undersaturated with concentration $c_A = 0.9c_{eq}^A$ , $p_{CO2}$ = 0.05 atm and $q = 0.05$ cm³/(node s). The water from below is undersaturated as well with $c_B = 0.9c_{eq}^B$ , $p_{CO2}$ = 1 atm and $q = 0.5$ cm³/(node s).

**Scenario B)** The meteoric water is undersatured $c_A = 0.9c_{eq}^A$ and the upwelling water is saturated, $c_B = c_{eq}^B$

**Scenario C)** Both, the surface water and the upwelling water are saturated with respect to calcite. This case is already illustrated in Fig. 4, right hand side panels and discussed there.

All other conditions in Scenario A and Scenario B are equal as in Fig. 4.

The temporal evolution of scenario A is illustrated in Fig. 13. In the beginning (panel a) we observe three regions of dissolution, the first at the water table where the aggressive surface water enters (yellow fringe). In the zone where rainwater mixes with the water from below renewed aggressivity arises by MC (yellow region). Finally, a plume extends from the input region of aggressive water from below (yellow). After 120 years (panel b) enhanced dissolution is observed in the bottom region where water from below enters and in the fringe of mixing (red). After 1300 years (panel c) dissolution at the bottom is restricted to

the input nodes. Mixing with high dissolution rates becomes restricted to a narrow band. Below this an extended zone of reduced rates (blue) is seen. Dissolution at the water table is similar to that of past times before. This pattern remains stable (panel d,e) until 30,000 years and thereafter. Most of the flow is focused to the water table cave and the cave system below. The black regions below show fractures with increased aperture widths invaded by saturated solutions. Therefore, dissolution in this region is absent.

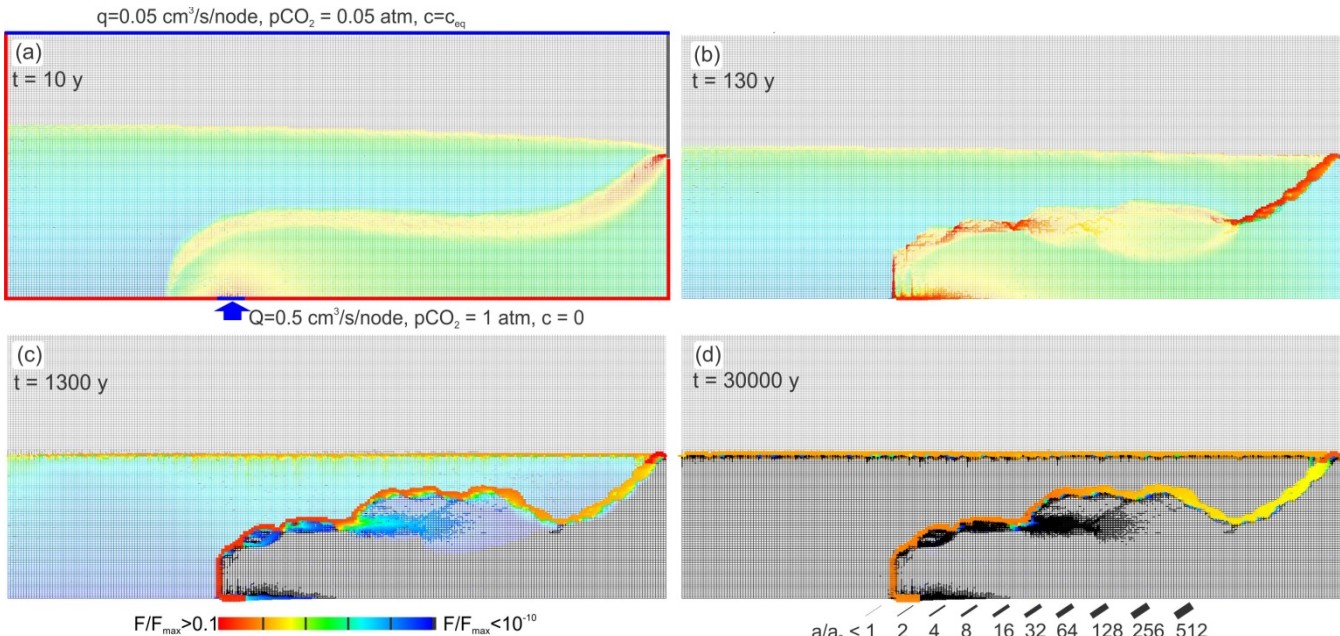


**Figure 13: Temporal evolution of scenario A. See text.**

Figure 14 shows evolution of Scenario B (saturated upwelling water, undersaturated meteoric input). In the beginning, two regions of dissolution are seen. The undersaturated rain water establishes a region of fracture widening at the water table.

Mixing of rainwater that has become almost saturated on its way down with the upwelling water creates a fringe of mixing corrosion. There a cave system is evolving. At the same time, the meteoric water widens the fractures at the water table creating a zone of high hydraulic conductivity, which in time drains most of the meteoric water to the output on the right. Therefore, the meteoric water does no longer intrude downwards and the mixing corrosion is no longer active. The entire region below

is invaded by the upwelling water that is saturated with respect to calcite ( black region) and further dissolution there stops. The water table cave however continues to grow.

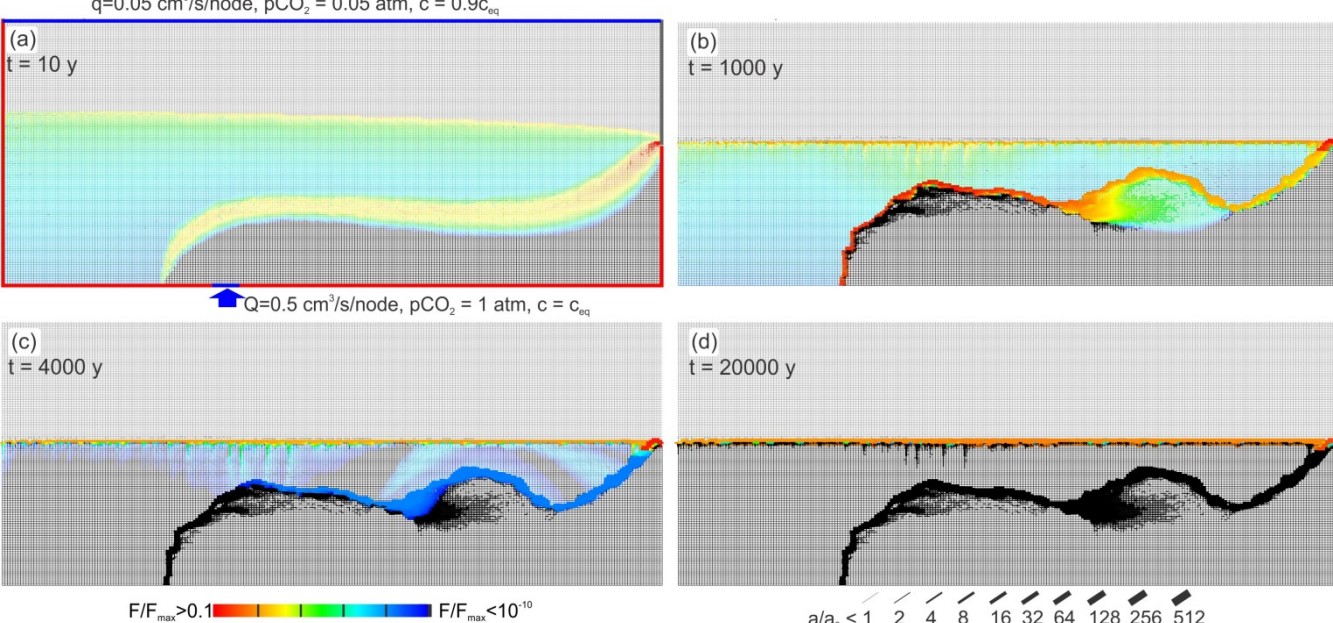

Figure 14: Temporal evolution of scenario B. See text

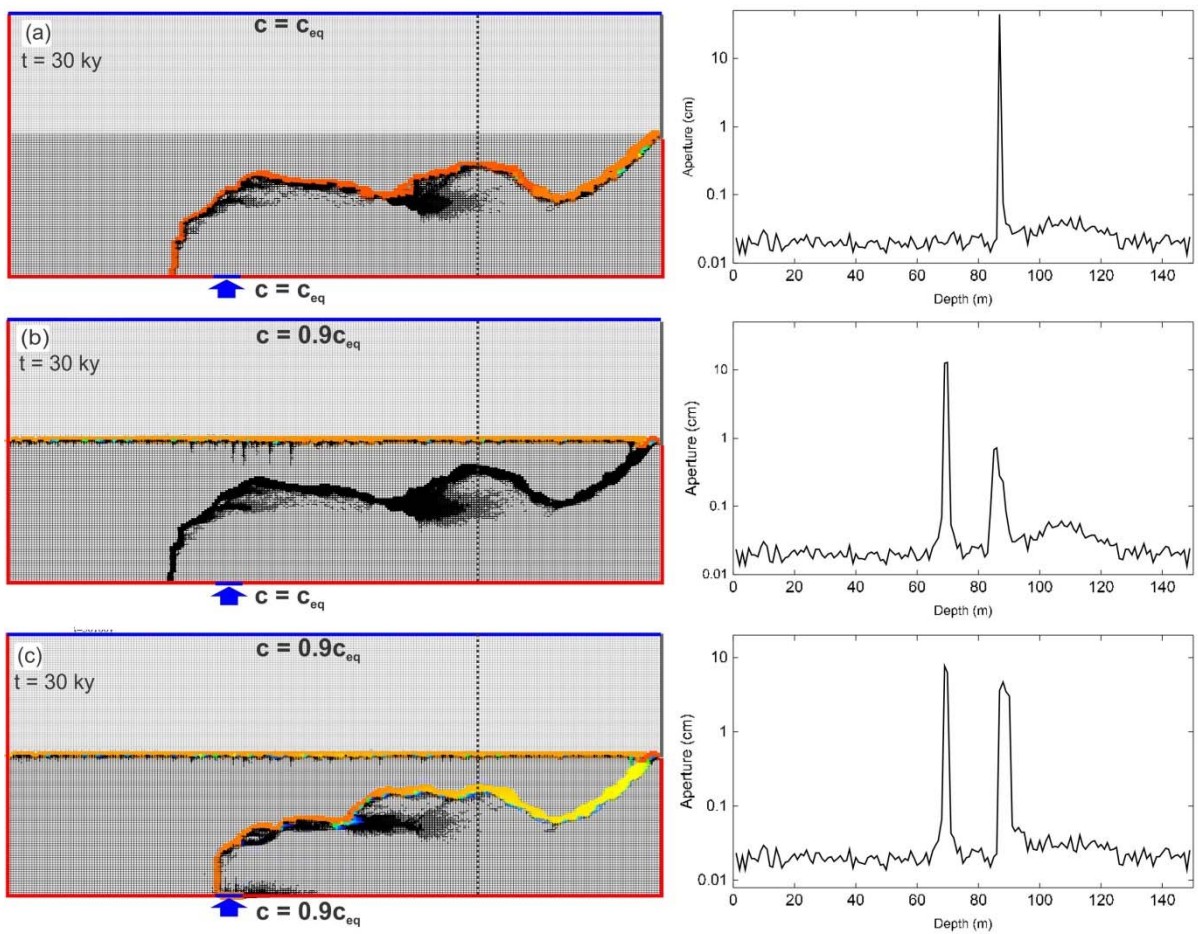


**Figure 15: Final evolution states of scenario A, Fig.13, scenario B, Fig. 14, and in Standard Scenario C, Fig. 4. Panels on the right show widths of horizontal fractures along profile lines shown in figures a-c.**

Figures 15 a,b,c illustrates the three final evolution states of scenarios A, B and C, presented in Fig.13. 14, and 4. Panel a

depicts the situation when both waters are undersaturated. Here due to the aggressive water from below dissolution rates along the mixture fringe are enhanced and flow from below occupies this region and that below it. Panel b depicts the final state when the meteoric water is aggressive only, as in Fig. 11. By the increasing hydraulic conductivity at the water table most of the flow of meteoric water is directed along this region. Mixing with the water from below is also restricted to this region where the water creates high dissolution rates.

In panel a, both, the meteoric water and the water from below are saturated with respect to calcite as in Fig. 4. Dissolution results from MC and lasts during the entire time of evolution. The right hand side panels 14 d, e, f illustrate the aperture widths of the horizontal fractures along the vertical transect as indicated in the corresponding left hand side.

## .8. Mixing Corrosion-fringe-instability in the mixing zone.

In all scenarios with pure MC discussed in this work we observe a common behavior during the early time of cave evolution. This is shown in Fig. 16. The left-hand side panels repeat the evolution already shown for the Standard Scenario in Fig. 4. At the beginning when dissolution has not yet been active, a wide fringe of mixed waters extends to both sides of the water divide between the two flow domains. At the output where flow is constricted to a narrow region the fringe of mixing is also narrow and one to one mixing occurs in its middle creating maximal widening of fractures as can be read from the red fractures. Therefore, flow is focused and the cave develops in this narrow region propagating upstream. This is a common feature in all scenarios where dissolution in a mixing fringe is of relevance.

After a short time the shape of the fringe is distorted (Fig. 16c). It is constricted to a narrow zone. In the middle of it dissolution rates are high as seen by red fractures. Beyond this constriction, an extended region of mixing with low rates of widening extends until it is squeezed again because its flow is channeled into the cave that propagates upstream.

The first constriction arises by an instability caused by chance due to the heterogeneity of the fracture aperture widths. Somewhere in the mixing zone, dissolution rates are maximal compared to the neighboring ones (Fig. 16b). They create a narrow region that attracts flow increasing further favorable mixing. This causes a feedback loop as already described. At the end of this region, flow is dispersed and the extended zone of mixing is created.

To show that it is the heterogeneity of the fracture widths that causes the instability we have investigated what happens (not shown here) when all fracture have equal uniform aperture widths of 0.02 cm. Here during the time ranges explored in Fig. 15 no instability occurred. This gives a warning in the reliability of digital models that are high idealizations of nature.

To give further support to our interpretation in the right hand side panels (Fig 16 d-f) we show the evolution of a scenario where a zone of wider fractures (0.04 cm) has been introduced as indicated by the black rectangle. In this case the constriction occurs exactly in this region.

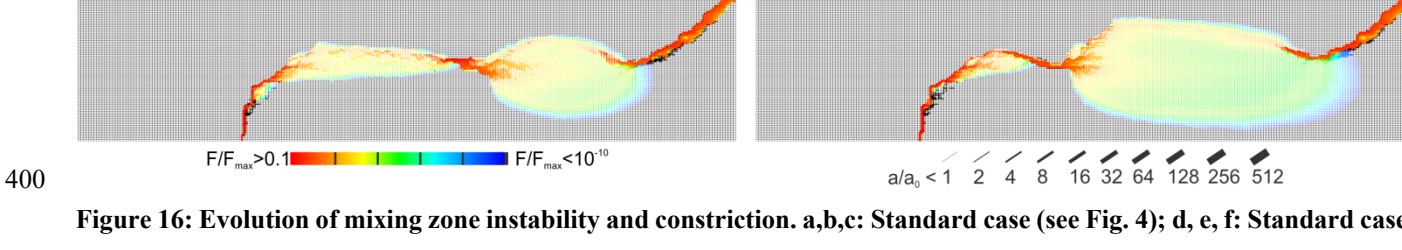

**Figure 16: Evolution of mixing zone instability and constriction. a,b,c: Standard case (see Fig. 4); d, e, f: Standard case, where instability was triggered by assigning initial aperture a0 = 0.04 cm to set of 4 horizontal fractures, marked by rectangle in panel d.**

One would expect that such constrictions should occur again in the downstream region. This, however, is prevented by focusing into the narrow output region. Therefore, we have extended the modelling domain from 500 m to 1400 m in the horizontal direction. The result is shown in Fig. 17. In the beginning, the fringe of mixing extends along the entire length of the domain. However, in the further evolution a series of constrictions arise as expected.

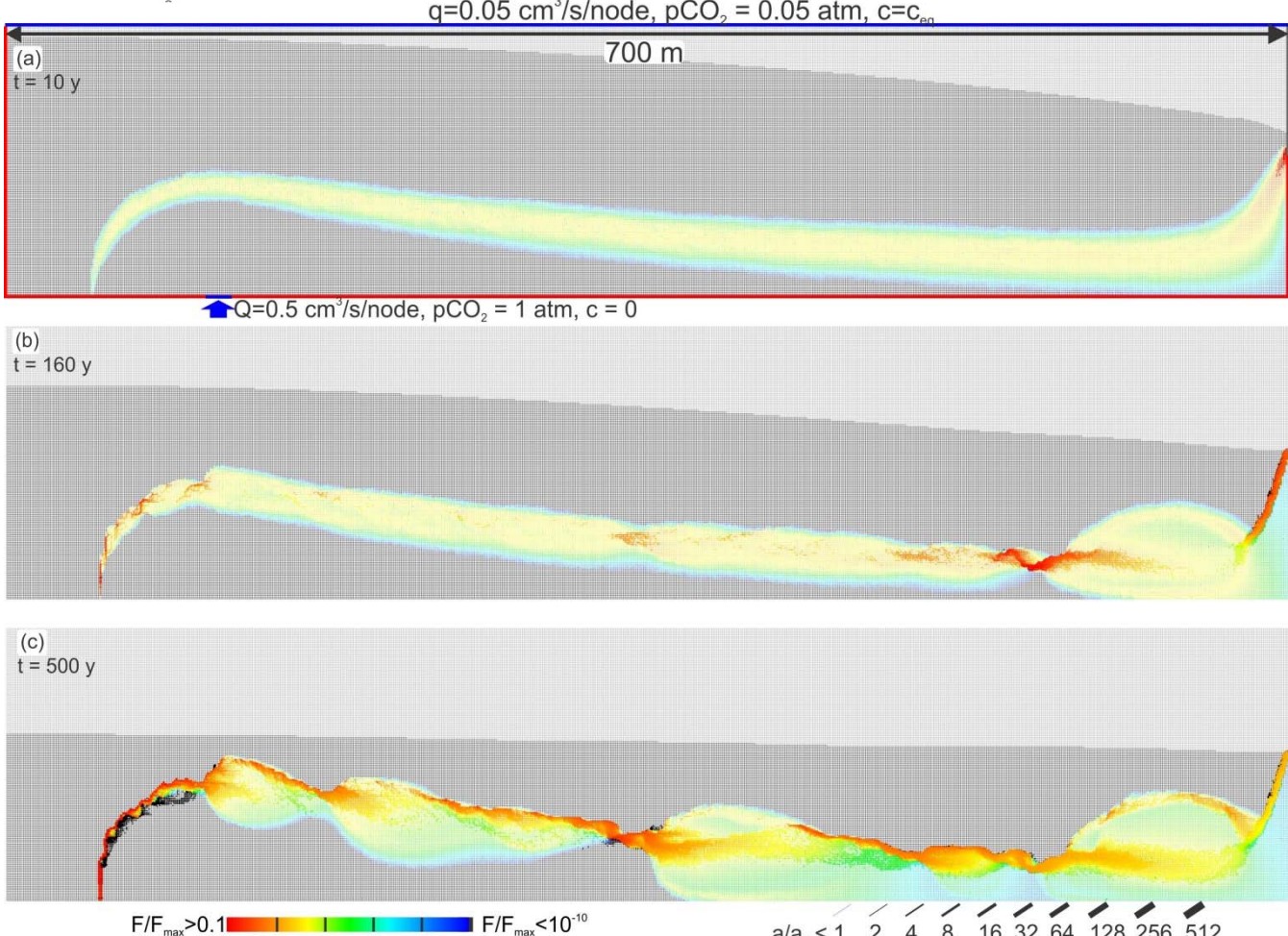

q=0.05 cm³/s/node, pCO₂ = 0.05 atm, c=c$_{eq}$

700 m

(a)
t = 10 y

Q=0.5 cm³/s/node, pCO₂ = 1 atm, c = 0

(b)
t = 160 y

(c)
t = 500 y

F/F$_{max}$>0.1   F/F$_{max}$<10$^{-10}$        a/a$_0$ < 1   2   4   8   16  32  64  128 256 512

**Figure 17: Standard scenario with horizontal extension of modelling domain from 500 m to 1400 m. Several constrictions occur downstream**

## 9. Impact of boundary conditions

In all our scenarios so far outflow was focused to the seepage face. To explore the impact of boundary conditions to the
evolution of karst by MC we have replaced the non-flow boundary below the seepage face by a constant head boundary with
head zero. In nature, this could be a river or a lake located at the rim of the rock massif that hosts the cave system.

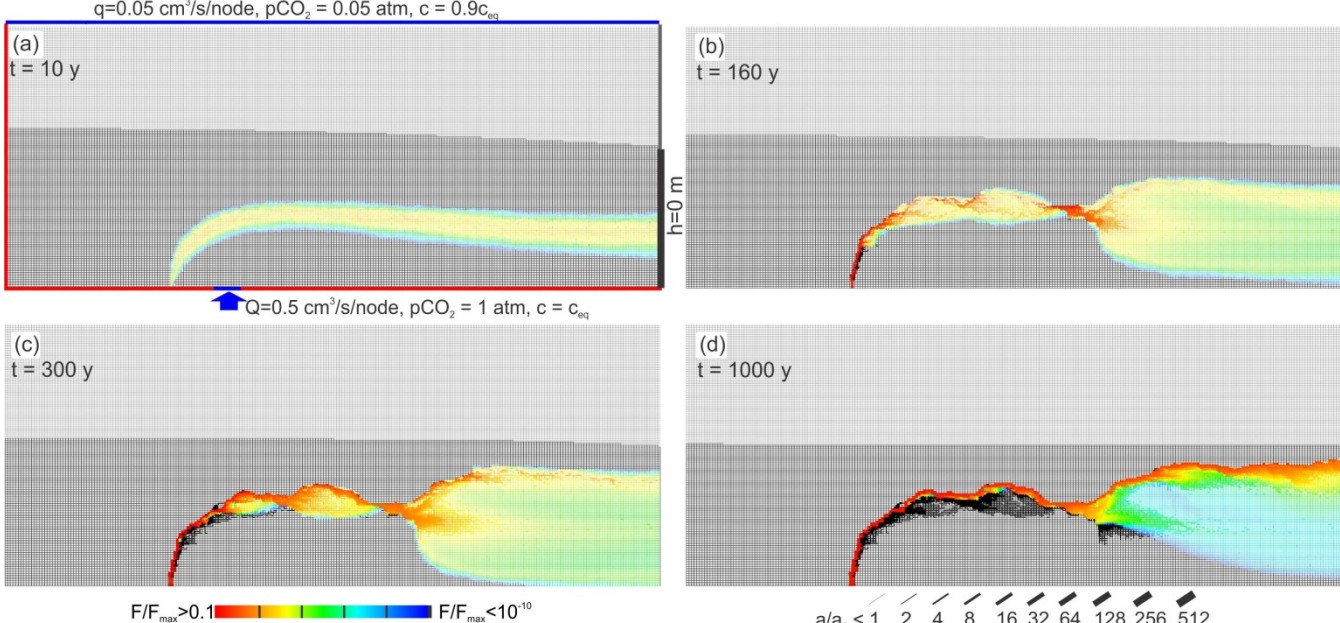

**Figure 18: Change of boundary conditions on the outflow to constant head conditions.**

The result is illustrated in Fig. 18. As in all scenarios that we have reported, a mixing fringe is present in the beginning. It
shows two constrictions downstream due to the MC-instability. From there a wide mixing zone extends with large widening
of fractures at its upper rim. These regions of high dissolution rates determine the location of the cave. This shows that the
MC-fringe-instability is a unique property of the MC-fringe independent of the specific output boundary conditions.

## 10. Pattern of flow

To further illustrate the origin of the mixing zone instability we have investigated the flow pattern and its relation to dispersive
mixing. Figure 19 presents the Standard Scenario with arrows presenting average direction and flow rate in 5x5 sub domains.
Length of the arrow presents the flow rate in a sub domain normalized to the flow rate in the sub domain with maximal flow.
This way flow lines can be envisaged easily.

Panel a) depicts the pattern of flow rates at the onset of dissolution. In the upper flow domain resulting from meteoric input all
arrows point downwards and all flow lines that can be envisaged from these arrows are first directed downwards, then they

bend and become almost parallel to the upper fringe of the mixing zone until they are directed to the outflow. There the flow lines are squeezed into a narrow band and accordingly their amount increases as can be seen by the long arrows. The flow lines inside the mixing zone and those touching it at its rims are almost parallel to each other. The flow lines in the lower flow domain originating from the hypogenic input rise from the input region then bend and become parallel to the lower fringe of

the mixing zone. Finally, they are directed towards the output.

Mixing between the surface water and the hypogenic water is possible only where the flow rate vectors are located in the mixing zone or close to the water divide. One by one mixing and accordingly, widening of fractures is enhanced in all regions in the mixing zone where the flow vectors have vertical components. These regions are indicated by shaded rectangles in panels a,b, c and d). From these one understands why dissolution rates are high at the output and at the region.


**Figure 19: Patterns of flow lines for the Standard Scenario.**

## 11. Discussion

By the action of mixing corrosion, hypogenic caves evolve in a large variety of geological settings. We have chosen our specific setup of an unconfined aquifer that is similar to the geology of the caves in the Black Hills (Palmer, 2017) and also bears a great similarity with that of hypogenic settings in the Guadalupe Mountains New Mexico, USA, e.g. Carlsbad Caverns, Lechuguilla Cave sculptured by sulfuric acid speleogenesis (SAS)(Palmer, 2006).

In SAS, oxygen rich waters from the surface mix with water rising from the depth that is saturated with respect to calcite and contains $H_2S$. This way similar as in CAS, a mixing zone is created. There by bacterial aid $H_2S$ is oxidized to $H_2SO_4$ that dissolves limestone readily thereby producing $CO_2$ as further agent for dissolution. This mixing process is close to that in our hypothetical CAS setup. However, there is not sufficient knowledge on details of this complex processes, especially about the location of bacteria in the fractures or their junctions and the kinetics of bacterial oxidation. Therefore, modelling of SAS is not possible. Nevertheless, the basic processes of cave formation should be similar to those of CAS. Therefore, albeit with care, our results may serve to interpret SAS cave systems.

In this work, we have focussed to a specific setting that offers new speleological mechanisms in an unconfined aquifer open to meteoric waters from the surface as they have been reported in the speleogenesis of the caves in the Black Hills, South Dakota, USA and the Guadalupe Mountains, New Mexico, USA. Such a model must be kept idealised and simple to reveal the basic mechanisms acting. It cannot explore the evolution of real cave systems because these may have undergone multiple phases of speleogenesis governed by different geological settings and hydrogeological conditions. Such changing conditions imprint the geological history to the cave evolution and would add a complexity that masks the basic contributions to further speleogenesis. The focus of this work is to construct a plausible hydrogeological setting to explore how and where fractures are enlarged by dissolution. This is a tool that reveals insights, which cannot be found otherwise. To our knowledge, no modelling of speleogenesis has ever been attempted to understand detailed patterns of real cave systems. Most modelling works can be regarded as building blocks to a better understanding of processes active in early karstification.

Of course, other scenarios can be envisaged. In our setting the aquifer initially may have been confined by impermeable strata on top of it. Then two possibilities exist. In the first the waters from below, saturated with respect to calcite are the only input of water. Then the aquifer is filled with that saturated water and karstification is not possible. It will start when the aquifer is unroofed by uplift and erosion. This is equivalent to our model. Here a limestone plateau is bordered by two valleys and due to the confining cover  the aquifer is filled with the upwelling water from depth avoiding dissolution of limestone until the aquifer is unroofed.

In the second possibility the aquifer receives water from some distant input upstream from the modelled region. Then karstification by mixing corrosion will inprint patterns of porosity and cave conduits that are inherited when the aquifer becomes unconfined. Although study of such a setup is beyond the scope of this work we give an example. We chose a setup that is similar to the one presented by Gabrovšek and Dreybrodt (2010).

Here we assume that water from a distant input flows into our aquifer that is now confined by impermeable strata on top of it. This is modelled by changing the boundary conditions of the standard case (see Fig. 4) to a constant input condition of saturated water with $p_{CO2} = 0.05$ atm the left hand side of the modelling domain and constant head at its right hand side.

Fig. 20 illustrates the result at three steps of the evolution. At the onset, two flow domains (red and blue) separated by a mixing fringe are clearly visible. After 500 years, a channel has developed from the input upwards and then horizontally. During its propagation, the solution from the channel is injected into the adjacent network, where it mixes with the regional flow. Dissolution in mixing zones imprints regions of increased permeability. At 500y we see regions of high permeability and active dissolution in the mixing zone that is restricted to the border of the two flow domains (red and blue in Fig. 20, d. e, and f). Such pattern then continues as the channel evolves towards the output boundary. This scenario is almost an upside-down image of the model presented by Gabrovšek and Dreybrodt (2010), where saturated solution from the surface mixes with the left-right regional flow.

To account for the fact that evolution of most aquifers started in confined settings, one could use end-member of such a model as an initial domain for the unconfined setting presented in this work. This could be an interesting topic for the future but is beyond the scope of this work that might be considered as a first step into this direction.

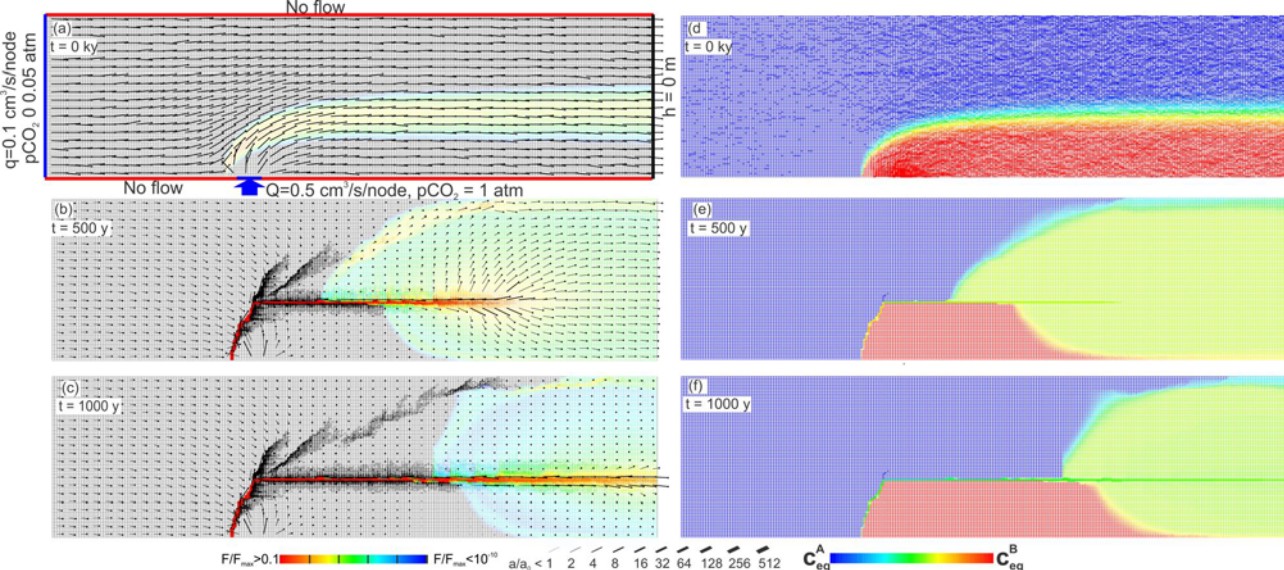

**Fig. 20: Evolution of the confined aquifer. The left panels represent fracture aperture widths (bar code below) and widening of the fractures in cm/year (colour code below). Black regions do not experience widening by dissolution. The right panels represent flow rates (bar code $Q/Q_{max}$ , where Q max is the maximal flow occurring in the net) and the values of $c_{eq}$ (colour code) of the water. Blue regions contain phreatic water from the input at the left-hand side of the modelling domain. Red regions are filled with upwelling water from below.**

Generally, one could argue that our setting is too idealistic. It is likely that a system of larger fractures intersects the idealised aquifer in our setting. Figure 21 shows a situation, where we have inserted a coarse grid of prominent fractures with aperture width of 0.05 cm. The deep flow is injected into a single prominent fracture as shown in Figure 21. Due to this injection, the

mixing zone forms similar as in the standard case (Fig. 4).The middle prominent fracture is located in this mixing zone and attracts flow as can be seen after 50 years by the red colour indicating high dissolution rates. Then flow focuses to the more permeable horizon of the prominent fracture and the mixing fringe is restricted to this region. However, as seen in Figure 21b, mixing zone constrictions also arise along the prominent fractures. The water from these constrictions is injected into the network causing additional growth as it mixes with the infiltrated water. This gives rise to growth of stable pathways within

the fine fracture network that bypass the middle prominent fracture (Figs. 21c and 21d).

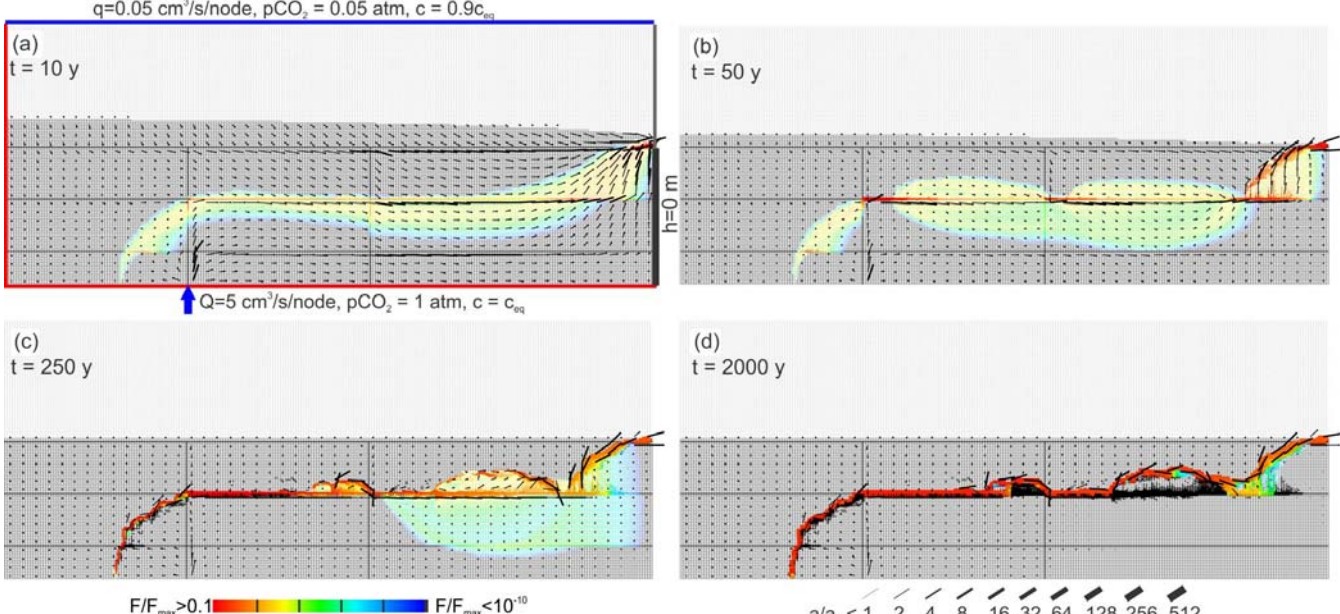

**Fig. 21: Regular network of prominent fractures of aperture width $A_0$=0.05 cm embedded into the standard network. Input of deep water is introduced along a single prominent fracture. Everything else as in standard scenario (Fig. 4).**

We do understand this behaviour in view of the insights we have gained from our idealistic setting. If we had explored the more complex setting from the beginning, insights revealed from the idealistic setting would have remained concealed. This is an example of the significance of idealistic simple models as building stones to understand more complex situations.

## 12. Conclusion

We have proposed first digital models of hypogenic carbonic acid speleogenesis (CAS) as suggested already in 1989 by Palmer and Palmer.They argued that where waters saturated with respect to calcite but with differing Ca-concentration are mixed mixing corrosion will become active as a cave forming mechanism. Many geological settings can be considered where such conditions are true. In this work we explore by digital modelling a simple idealized setting. Meteoric water percolates into a limestone massif attaining saturation with respect to calcite when it meets the water table. From below deep-seated water

loaded with $CO_2$ rises reaches saturation with respect to calcite and invades into the limestone. In Darcy flow two flow domains are established that are separated by a water divide. There, due to dispersive diffusion, the different waters mix in a fringe surrounding the water divide and mixing corrosion sets in widening the aperture of the fractures. The water leaves the aquifer at a seepage face in most of our computer runs. The cave evolves first at the output and in the depth of the aquifer close to the input from below. Between these regions a wide fringe of smaller dissolution rates develops.

This fringe in its middle due to the heterogeneity of the fractures develops constrictions with increasing dissolution rates and attracts flow from both waters that mix there to reach maximal dissolution rates. Finally a cave system bending up and down attracts all flow such that its pattern stays stable in time but with increasing widths of conduits.

Depending on the parameters of the computer runs such as e.g. dimensions of the aquifer, magnitude of water input, q, from below and the amount, $Q$, of meteoric precipitation as well as the ratio $Q/q$ that determines the position of the initial flow

domains a variety of cave patterns are obtained. We have modeled cases when the meteoric input varies in time and found that the early state of cave evolution determines its final pattern.

Our modelling results have some consequences to the definition of hypogenic karst. Both definitions, that by Palmer (2000) from the geochemical view as well as that by Klimchouk (2007;2016) from a hydrological approach state independence of hypogenic karstification on surface processes. In CAS as well as in SAS, however, surface processes do have a strong impact.

## 13. Author Contributions


WD initiated the work and wrote the text. FG is the author of the model code. He performed simulations and prepared figures. The manuscript is based on the in-depth discussions of both authors.

## 14. Competing interests

The authors declare that they have no conflict of interest.

## 15. Acknowledgments


FG acknowledges the financial support from the Slovenian Research Agency (research core funding No. P6-0119).

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
