# Peer review of "Early hypogenic Carbonic Acid Speleogenesis in unconfined limestone aquifers by upwelling deep-seated waters with high CO2 concentration: A modelling approach"

_Hydrology and Earth System Sciences, 2020_

## Referee Comment (RC1) · Steffen Birk (Referee) · 13 Nov 2020

This modelling study explores the evolution of caves in limestone settings where meteoric waters mix with upwelling deep-seated waters with high pCO2. Most of the scenarios shown assume that both the meteoric and the deep-seated waters are saturated with respect calcite such that the development of caves only results from mixing corrosion in the zone where the two flow components mix. In addition, scenarios are

considered where one or both of the flow components are undersaturated with respect to calcite.

The scenarios shown in the paper give insight into the mechanisms of speleogenesis within this particular type of setting. Thus, the paper addresses relevant scientific questions within the scope of HESS. The scenarios shown are interesting and novel. However, it is difficult to assess how far the general concepts and ideas go beyond those that have been presented by similar modelling papers, as there is almost no comparison made to other modelling studies. It seems to me that this should be more clearly addressed in the introduction of the paper and in the discussion of the results by referring to the scientific literature. In particular, the discussion is very short and does not refer to other modelling studies, but it is also not very clear from the introduction how this contribution classifies into the existing (modelling) studies and the two different definitions of hypogene speleogenesis. I therefore recommend revision particularly of the introduction and discussion section.

Specific comments:

1) Abstract, l. 9: "Hypogene caves originate from upwelling deep-seated waters loaded with CO2 that mix with meteoric waters . . ." – in this general form this statement is not valid. Even if one applies the "geochemical view" of hypogene speleogenesis as outline in l. 39-41, it does not require mixing with meteoric waters, but any "aggressiveness [. . .] produced at depth" would be sufficient. Thus, this sentence needs to be revised to clarify that this paper considers one specific type of hypogene setting or at least indicate that this is only one type of hypogene speleogenesis (e.g. "Hypogene caves may originate from . . .").

2) Abstract, l. 18: Please correct "seepaall ge face" (seepage face?).

3) Introduction, l. 50/51: "In this work we take a first step to explore by digital modelling the processes governing the initial evolution of hypogene caves. " Again, this general statement is not valid. This appears to be related to the statement addressed by comment 1, which suggests that the specific scenario considered in this paper generally represents hypogene speleogenesis. As pointed out in comment 1, however, even if the geochemical definition of hypogene caves is used there are other mechanisms that produce aggressiveness at depth, such as the temperature effect mentioned in l. 45-47. Therefore, at least some of the papers cited in those lines classify as models of hypogene speleogenesis. The cited paper by Chaudhuri et al. (2013) even is titled "Early-stage hypogene karstification . . .", and the cited paper by Rajaram et al. (2009) includes a section titled "Hypogene karst simulations". I think there are other, similar papers that are relevant, such as Andre and Rajaram (2005): Dissolution of limestone fractures by cooling waters: early development of hypogene karst systems. Water Resour. Res., 41, W01015. There are even more modelling studies addressing hypogene caves if one adopts the "hydrological" definition given in l. 41-44. In particular, the two cited papers by Birk et al. (2003) and Li et al. (2020), which consider the artesian hypogene speleogenesis as observed in the Western Ukraine, belong to this category, and again there are other papers addressing this type of setting such as Birk et al. (2005): Simulation of the development of gypsum maze caves. Environmental Geology 48 (3): 296-306; Rehrl et al. (2008): Conduit evolution in deep-seated settings: Conceptual and numerical models based on field observations. Water Resources Research 44, W11425; Rehrl et al. (2010): Influence of aperture variability on conduit evolution in hypogene settings. Zeitschrift für Geomorphologie, Suppl. 54 (2): 237-258.

4) In the light of the previous comment, l. 48-50 (citing Klimchouk et al. 2017) also appear to be inappropriate, in particular, as Klimchouk (2013), which is also cited in the paper, does refer to some of the above-mentioned modelling studies of artesian hypogene speleogenesis.

5) It is further noteworthy that with the "hydrological approach" (l. 41) the scenario considered here does not classify at all as hypogene speleogenesis, because the underlying mechanism clearly is not "independent of direct recharge from the overlying or immediately adjacent surface" (l. 43). This, of course, is not problematic, as the geochemical definition is fully acceptable. Nevertheless, I think that the authors should be clear about this, e.g. by explicitly saying that this work adopts the geochemical definition. Somehow the discussion of "agents active in hypogene speleogenesis" in l. 53-65 hints at this, but it could be introduced by a statement that explains this more clearly. I think that this would also make it easier to explain, e.g. at the end of the introduction, how this work goes beyond the existing research – as I understand, it focuses on hypogene settings in a geochemical view, and within those settings it addresses chemical mixing corrosion (as opposed to the temperature effect considered by others), etc.

6) Even if the focus (and thus the novelty) of the paper is more precisely defined as suggested in the previous comment, I wonder if there is other related work that should be addressed in the introduction and even more importantly in the discussion of the paper. One the one hand, this could be papers more generally addressing the role of mixing corrosion in cave evolution, particularly the paper by the same authors published in 2010 ("Karstification in unconfined limestone aquifers by mixing of phreatic water with surface"), which appears to address similar processes in a setting that differs in the source of one of the flow components but still seems sufficiently similar to warrant a comparison of the results and a discussion about the differences resulting from the different type of setting. On the other hand, there is a recent paper also addressing the role of mixing of meteoric water with "cross-formational warm water" (Gong et al. 2019: Modelling early karstification in future limestone geothermal reservoirs by mixing of meteoric water with cross-formational warm water. Geothermics 77:313-326), i.e. the paper appears to address a similar type of setting albeit with a focus on other (thermal) processes. It would be very interesting to learn about the similarities and differences between the results from simulations with this type of settings/processes and studies that looked at either at similar setting (but different processes) or similar processes (but different settings). For instance, it would be very interesting to see if there are features of the resulting cave patterns that are characteristic only for the setting/processes considered here. This and similar aspects should be addressed in the discussion section.

7) There seem to be some omissions in the list of references. Palmer (2017) is cited in l. 445, but I cannot find a corresponding publication in the reference list. The same applies to Palmer and Palmer (1989) (l. 454/455). Please check specifically these two references and also more generally the completeness of the reference list.

8) The paper includes a fairly high number of figures, obviously because several scenarios are described and discussed. I think all of the scenarios are generally of interest, but I wonder if all of the figures are needed. Perhaps not all of the figures showing the aperture widths along profiles are needed? I find it difficult to give clear recommendations in this regard, but I think the authors might want to think about this issue once they have more clearly worked out, which aspects of their models are most relevant when they compare with the findings from related modelling studies (as suggested above, particularly in comment 6). Yet, I want to emphasize that each of the figures is of interest and thus in my view deserves to be shown if the authors think they are needed to illustrate important aspects of their results.

---

## Referee Comment (RC2) · Alexander Klimchouk (Referee) · 25 Nov 2020

This study simulates the development of solution porosity in specific settings where unconfined limestone aquifer recharged from the surface also receives the localized input of CO2-rich ascending deep waters at the bottom. The initial permeability structure in the highly idealized aquifer is represented by a rectangular net of fractures with apertures selected from a truncated log-normal distribution. The study focuses on

scenarios where both waters (that from the surface recharge and that from the deep source) are saturated with respect to calcite when they interact in the phreatic zone and dissolution is caused by their mixing, although scenarios are also modeled where one or both waters retain the aggressiveness.

Modeling hypogene karstification (speleogenesis) is highly relevant as its regularities and peculiarities are much less studied than those of more familiar epigene karst. This is especially true considering the great variability of settings and processes of hypogene karstification. This study models speleogenesis in particular settings (as outlined above) by the particular processes (mixing dissolution by carbonic acid) and provides insight into the regularities and peculiarities of speleogenesis in the modeled situation. The modeling part of the work is excellently realized for the chosen settings and conditions.

There are, however, some questions regarding the general presentation of hypogene karstification (Comments 1a-1c below), proper appraisal of previous modeling studies (Comment 2), and how representative is the modeled setting for hypogene karstification (Comment 3a-3c).

1a. Some statements regarding hypogene karst are inadequate and misleading due to the overgeneralization of specific situations. The authors state in Abstract (lines 9-10) that "Hypogene caves originate from upwelling deep-seated waters loaded with CO2 that mix with meteoric waters in a limestone aquifer." This, however, defines only one specific mechanism and situation of hypogene speleogenesis, modeled in this study. It is neglected that hypogene caves originate from a number of processes, including those in which CO2 and mixing are not involved, and in different lithologies, not only in limestones.

1b. In line 54, the authors open the listing of "various agents active in hypogene speleogenesis" but the subsequent list includes only two of those considered in the relevant literature: (A) dissolution of limestones by sulfuric acid (aka sulfuric acid speleogenesis

– SAS) and (B) dissolution of limestones by carbonic acid (aka carbonic acid speleogenesis – CAS). This listing ignores other agents active in hypogene speleogenesis, for instance in evaporite rocks or in silicate rocks, and therefore it should not be introduced in this general form.

1c. It is argued in Discussion (line 446) that "CAS is similar to sulfuric acid speleogenesis (SAS)". In the subsequent paragraph, this statement is reasoned by that SAS and CAS similarly develop in unconfined aquifers from mixing of water from the surface with water rising from the depth. This, again, is an inappropriate generalization of the modeled situation to carbonic acid speleogenesis in general. For instance, carbonic acid speleogenesis in deep settings due to retrograde solubility of calcite in rising and cooling $CO_2$-rich thermal waters (aka hydrothermal speleogenesis) is regarded to be one of the most common types of hypogene speleogenesis but it is in no way similar to SAS.

2. In several places in the text, this study is inappropriately presented as the first step in numerical modeling of hypogene speleogenesis whereas a number of works exist where it has been modeled in other settings and with other chemical processes (see references in the S.Birk's comment). At least one study (Gong et al., 2019) has modeled hypogene karstification in settings similar to that explored in this work. Thus such general claims of priority must be avoided and appropriate citation of other modeling works is needed when the authors refer to hypogene speleogenesis in general (lines 10, 50-51, 459). I agree with the other referee (S.Birk) that comparison with other modeling studies and more discussion of other related works should be recommended.

3. How realistic and common in natural conditions is the modeled situation? Some discussion is needed why this particular modeling domain and setting are chosen to model hypogene karstification and how representative is the modeled situation. In my opinion, it seems to be of rather limited importance in nature. Some of the reasons for this appraisal are outlined below.

3a. The input of the deep waters from below is designed in this study as a region (a "window") at the otherwise impermeable bottom, through which this rising water enters in a dispersed way into numerous "common" fractures in the limestone, forming the separate flow domain in the unconfined aquifer. In reality, a much more common mode of the input of deep waters into a shallower aquifer is through highly localized cross-formational tectonic disruptions such as faults and large fractures (so called "through-going" fractures) that cross both, the confining unit at the bottom and the shallower aquifer. The deep water remains largely canalized in a large fracture during further ascent and mixing occurs in and around it through the interaction with flow in the net of "common" fractures.

3b. From the perspective of geological evolution, a limestone aquifer commonly becomes unconfined as the result of uplift and denudation of the stratified formation, i.e. through the removal of the upper confining unit. This means that the now unconfined aquifer was once part of a confined aquifer system in which localized cross-formational hydraulic communication occurred. Such vertical communication (and hence the ascending input into the given aquifer) commonly occurs during long time spans on the way of the given formation from burial to the shallow subsurface and is particularly intensified when the upper confining unit is getting thinner and eventually locally breached. Thus, the inputs of the deep waters from below are usually inherited in the active state from the confined situation but not open after unconfined conditions have established.

The expected result would be (using the same chemical properties of two waters - the aquifer water and the rising deep water) that prior to the complete unroofing the given aquifer has experienced some hypogene karstification due to mixing dissolution so that substantial heterogeneity of the permeability structure would be present. In other words, the initial condition of a non-karstified aquifer that once starts receiving the localized input of the deep waters at the bottom seems to be unrealistic. It is worth to mention that the authors shown through their modeling exercises the great role of

initial heterogeneities in the development of karstification patterns and emphasized as the finding that "the early state of cave evolution determines its final pattern" (line 470).

There may exist unconfined limestone aquifers that have not experienced burial yet, e.g. eogenetic limestones in young carbonate platforms. The modeled domain and setting may be relevant to this situation although (1) the fracture networks in eogenetic carbonates are commonly less regular and certainly differ from that inserted in the model, and (2) syngenetic karstification is likely to create considerable heterogeneity in the beds which, upon burial beneath younger beds, host the phreatic zone of the modeling domain.

3c. Two interacting waters (the aquifer water and the rising deep water) are taken in this study as of equal temperature and density, which is not common in such situations. The density differences between the two sources, if accounted for, would certainly have an impact on the flow pattern, mixing, and karstification.

In summary, this study provides insight into the mechanism of karstification in the chosen type of settings. However, revision is recommended with regard to the following aspects: (1) more adequate general presentation of hypogene karstification and clear acknowledgment that this study characterizes only one specific situation of it; (2) recognition of other modeling studies of hypogene speleogenesis; (3) discussion explaining why this particular setting is chosen to model hypogene karstification and how representative is the modeled situation.

---

## Author Comment (AC1) · 27 Dec 2020

**AUTHOR'S REPLY**

We would like to thank Steffen Birk for the helpful review. We have considered his comments and critics. Below find our replies to general and specific comments. We also propose changes in the manuscript.

Several issues have been addressed by both reviewers. To this extend part of replies to both reviewers overlap.

**General comment:** The main objective of both reviewers is that we generalize our special case of Carbonic Acid Speleogenesis. We must stress that we are aware of the other modelling efforts on the hypogene speleogenesis. Given the two quoted conceptual frameworks (hydrological and geochemical), speleogenetic settings are extremely diverse. All aspects are therefore beyond the scope of our work; however we agree that a better description of how our settings fit within the framework of the conceptual models, is needed.

To do this, we will reformulate the introduction and discussion to clarify that our model is a specific case in a wide variety of hypogene settings. Other modelling studies will be mentioned and properly cited.

The reviewer then specifies the comments in 8 points. Below we give a list of replies and intended changes in the manuscript:

**Point 1: Abstract:** We agree with the critics. To address it, the first sentence in the abstract "*Hypogene caves originate from upwelling deep-seated waters loaded with $CO_2$ that mix with meteoric waters in a limestone aquifer.*" will be replaced by "Here *we present first results on digital modelling of a specific setting of hypogene Carbonic Acid Speleogenesis (CAS)*." The last sentence will be changed to "*These findings give important insight into mechanisms of carbonic acid* speleogenesis *(CAS) in a special setting of unconfined aquifers. They also have implications to the understanding of corresponding sulphuric acid speleogenesis (SAS).*"

This clarifies that our paper considers a specific type of hypogene setting and speleogenesis. However, the mechanisms discovered with our model may be applied to other hypogene settings.

**Point 2: Abstract:** The typo will be corrected.

**Point 3: Introduction:**

We mostly agree with the comment. To this extend the introduction will be rewritten in order to put our settings within the framework of the two general concepts proposed by Klimchouk and Palmer. The hydrological concept of Klimchouk will be first introduced. We will refer to earlier modelling efforts that explore this concept in terms of "thermal hypgene speleogenesis" and cross-formational upwelling flow. The introduction to the geochemical concept will be extended. We will clarify that our work refers to an idealistic specific hypothetic setting of an unconfined aquifer for both SAS and CAS.

**Point 4:** Is considered in reply to P3.

**Point 5:** Thank you for pointing this out. The point can easily be addressed in the introduction. We agree that the settings presented in our work apply to the geochemical concept, which in principle needs mixing of waters with different origin to make the dissolution possible. To avoid confusion, this will be clearly stated in the introduction. By the way we have stated this already at the end of the conclusion:" *Our modelling results have some consequences to the definition of hypogene karst. Both definitions, that by Palmer (2000) from the geochemical view as well as that by Klimchouk (2007;2016) from a hydrological approach state independence of hypogene karstification on surface processes. In CAS as well as in SAS, however, surface processes can have a strong impact.*

**Point 6:** A direct comparison with other modelling studies and discussion of other related work as requested by both reviewers is not possible because such papers do not exist in the literature. The work of Gong *et al*. uses a modelling domain similar to ours. But it deals with thermal uprising water that gains renewed aggressiveness by cooling on its way up. As heat transport here plays a crucial role as driving mechanism a comparison to our work is not possible.

We suggest to present two new scenarios in the revised version: The first scenario is a response to Steffen Birk's comment referring to our work published in 2010 ("Karstification in unconfined limestone aquifers by mixing of phreatic water with surface"). In this scenario we assume that water from a distant input flows into the aquifer that is now confined by impermeable strata on top of it, and upwelling water enters into that aquifer from below. It turns out that here different processes act because a water table does not exist in this setting. From the results of these scenarios, we show that modeling must start from simple idealized settings to understand situations that are more complex.

**Point 7:** Will be corrected.

**Point 8:** We agree that the work is somehow graphical. We have already limited the number of cases to those that clearly present the basic findings. Although the aperture profiles can be estimated from the panels, they provide better "metric" information on the size of the conduits. We therefore in view of the on-line edition prefer to keep the figures.

---

## Author Comment (AC2) · 27 Dec 2020

**RESPONSE TO THE REVIEW OF ALEXANDER KLIMCHOUK**

We thank to Alexander Klimchouk for the helpful review. We have considered these comments and critics. In this reply we also describe planned changes in the revised manuscript.

The main concern of both reviewers is that our work addresses specific settings of hypogene speleogenesis and that this should be stated clearly. Several other issues have also been addressed by both reviewers. Therefore some parts of our replies to both reviewers overlap.

**General comment:**

The main objective of both reviewers is that we generalize our special case of Carbonic Acid Speleogenesis. We must stress that we are aware of other modeling efforts on hypogene speleogenesis. The two quoted conceptual frameworks (hydrological and geochemical), speleogenetic settings are extremely diverse. All aspects therefore cannot be addressed. However, we agree that a better description is needed how our settings fit within the framework of general conceptual models.

To do this, we will reformulate the introduction and discussion to clarify that our model is a specific case in the wide variety of hypogene settings. Other modelling studies will be mentioned and properly cited.

We therefore suggest changes in the abstract, introduction and discussion.

The reviewer specifies his comment in three sections (1a-1c, 2, 3a-3c).

**Comment 1a:**

This has been raised by both reviewers and we agree with the critics. To address it, the first sentence in the abstract "*Hypogene caves originate from upwelling deep-seated waters loaded with $CO_2$ that mix with meteoric waters in a limestone aquifer.*" will be replaced by " *Here we present first results on digital modelling of a specific setting of hypogene Carbonic Acid Speleogenesis (CAS).*" The last sentence will be changed to "*These findings give important insight into mechanisms of carbonic acid speleogenesis (CAS) in a special setting of unconfined aquifers. They also have implications to the understanding of corresponding sulphuric acid speleogenesis (SAS).*"

This clarifies that our paper considers a specific type of hypogene setting and speleogenesis.

**Comment 1b and 1c:**

To address the concerns raised, the introduction will be rewritten to place our setting within the framework of the two general concepts proposed by Klimchouk and Palmer.

The hydrological concept of Klimchouk, will be first introduced. The modelling works, which explore this concept in terms of "thermal hypgene speleogenesis" and cross-formational upwelling flow will be presented to give a broader overview. The introduction to the geochemical concept will be extended.

**To conclude**: we agree that our concept is not general but a specific case of hypgene speleogenesis. The Introduction will be changed to present it from the perspective of other more general concepts. We will clarify that our work refers to a specific hypothetic setting of an unconfined aquifer for both SAS and CAS.

**Comment 1.c:** Is considered in reply to 1b.

**Comment 2:** We are aware of earlier model of hypogene speleogenesis. However, a direct comparison with other modelling studies and discussion of other related work as requested by both reviewers is not possible because other works focus to different processes. The work of Gong et al. uses a modelling domain similar to ours. But it deals with thermal uprising water that gains renewed aggressiveness by cooling on its way up. As heat transport plays a crucial role as driving mechanism a comparison to our work is not possible. We will mention the work of Gong in the introduction to give a further example of multiple hypogene situations. We will take care that the reader is aware that out model is not a general one.

In summary: To meet the concerns raised in Comment 1 and 2, we will reformulate the introduction and discussion to clarify that we do not model general hypogene speleogenesis but a specific case. Other modelling studies will be mentioned by adding literature.

**Comment 3a:**

To study basic mechanisms we have chosen a simple scenario as most of the modelers do to reveal basic mechanisms. As an example, to apply multiple inputs from below adds complications and may hide the basics. Models cannot describe real caves, but in highly idealized settings they are a tool to understand processes acting in the early evolution of caves. Similarly, single fracture is a modeling artifact in epigene speleogenesis, but it is at the same time a basic building block for understanding epigene speleogenesis.

**Comment 3b:**

We agree that the unconfined setting with deep water inflow may be a special case, although relevant for situations where upwelling regional flows mix with local autogenic recharge. It is obvious that the imprint of past evolution (as for example in confined settings) makes a critical imprint to initial settings of our scenario. This, however, would mask the basic contributions to speleogenesis revealed by our idealised simple setting.

We do not agree that our unconfined setting is unrealistic. It may well be possible that permeable rocks cover the limestone strata thus allowing connection to the surface water. It may also be possible that the confined aquifer does not experience karstification that starts only when the confining cover is removed. Such situations cannot be excluded. It is, however, beyond the scope of our paper to discuss all these details raised in 3b. Actually these concerns question most modeling efforts so far published.

With respect to the geological evolution of an initially confined aquifer we have pointed out that the geological history of the cave evolution would add a complexity that masks the basic contributions to speleogenesis revealed by our idealised simple setting.

Furthermore we have added two new scenarios: The first is similar to our work published in 2010 ("Karstification in unconfined limestone aquifers by mixing of phreatic water with surface" as requested by S. B. Here we assume that water from a distant input flows into our aquifer that is now confined by impermeable strata on top of it and upwelling water enters into that aquifer from below. We have

discussed the evolution of this setting by adding a new figure 20. It turns out that here different processes act because a water table does not exist in this setting.

We will add a second new scenario where a simple net of wide fractures is inserted into our net of relatively homogeneous fractures. Therefore we have to add another figure. At that point we want to state that we regard all figures in this work as useful and necessary. With respect of the online version of the paper this should not be a problem. From the results of this scenario we will show that modeling must start from simple idealized settings to understand more complex situations. We consider this as an example of the significance of idealistic simple models as building blocks to understand more complex situations.

**Comment 3c:**

Yes, it is likely that the temperature of the upwelling water differs from that of the infiltrating meteoric water. This adds complexity to the model as heat transport coupled to density driven flow needs to be considered as well as temperature dependence of calcite solubility. To include that into our model is not possible. One may question therefore the modelling efforts in general. We do not know any model in the literature that is complete with respect to other mechanisms that could act.

**To conclude:**

We are aware that this model addresses a limited set of situations possible in nature and that it is far from being general. On the other hand we would like to stress that aim of such modelling is to explore new mechanisms which could be important in natural settings. Such mechanisms are the basic building blocks of our understanding of more complex settings. By introducing all the natural complexity at once, these mechanisms cannot be seen. Of course, the complexity could be gradually added.

The model presented here demonstrates how the imprint of the early pathways may define the long-term evolution of conduits and therefore the structure of the resulting aquifer. It also demonstrates how flow focusing provokes mixing of contrasting solutions and sustains the high dissolution rates in the conduits. Focusing may be triggered by small irregularities or it can be a consequence of boundary settings.

**Summary:**

To summarize, we have revised the text to meet points 1 and 2 of A. K's summary, especially by rewriting the introduction and discussion to clarify that we do not model generally hypogene speleogenesis but a specific case. We have recognized other modeling studies by adding literature as suggested (point 2).

We have also introduced the scenarios suggested by A. K. Furthermore we have pointed the limitations of all attempts to model early karstification and explained why such models can reveal processes but cannot represent real karst systems (point 3). In other words: Models are a good tool to understand karstification but a bad tool to predict it.

---

## Author Response (AR1)

**Authors' Responses to Reviews**

The separate responds to comments of both reviewers were uploaded to the Interactive discussion.

https://hess.copernicus.org/preprints/hess-2020-473/hess-2020-473-AC1-supplement.pdf

https://hess.copernicus.org/preprints/hess-2020-473/hess-2020-473-AC2-supplement.pdf

Here, we combine the responses, and point to corresponding changes in the manuscript. We have also considered the editor's remarks.

**We have uploaded a version with tracked changes and a clean version of the manuscript. The new references are marked red in the version with tracked changes.**

**General comments:**

The main objective of both reviewers is that we generalize our special case of Carbonic Acid Speleogenesis. We must stress that we are aware of the other modelling efforts on the hypogene speleogenesis. Given the two quoted conceptual frameworks (hydrological and geochemical), speleogenetic settings are extremely diverse. All aspects are therefore beyond the scope of our work; however we agree that a better description of how our settings fit within the framework of the conceptual models, is needed. The changes in the Introduction and Discussion were made to consider these comments.

**Terminology issue:**

We have also changed the term hypogene to hypogenic in the entire the manuscript. This was suggested by a native speaker that we discussed with (A.N. Palmer) who suggest to use *"hypogenic" or "hypogenetic"*, because *"gene" is a physical object, while "--genic" or "genetic" refers to origin.*

**Changes in the manuscript:**

**Title**

In the title, we have deleted the word *first,* so that it reads "Early hypogenic Carbonic Acid Speleogenesis in unconfined limestone aquifers by upwelling deep-seated waters with high $CO_2$ concentration: A modelling approach"

**Abstract**

We have considered the comments (1&2) by Steffen Birk (SB) regarding the abstract. We have replaced the first sentence of the abstract "*Hypogene caves originate from upwelling deep-seated waters loaded with $CO_2$ that mix with meteoric waters in a limestone aquifer."* by "Here *we present first results on digital modelling of a specific setting of hypogenic Carbonic Acid Speleogenesis (CAS).*" We have also changed the last sentence to "*These findings give important insight into mechanisms of carbonic acid speleogenesis (CAS) in a special setting of unconfined aquifers. They also have implications to the understanding of corresponding sulphuric acid speleogenesis (SAS)."*

This clarifies that our paper considers a specific type of hypogenic setting and speleogenesis. However, the mechanisms discovered by our model may be applied to other hypogenic settings. These changes also consider general comments given by Alexander Klimchouk (AK).

**Introduction**

We have considered the comments 3-5 by SB and comments 1&2 by AK as pointed out in our replies to the reviews. To this extend the Introduction has been largely rewritten. We have placed our scenario within general framework of hypogenic speleogenesis and models of s as proposed by Palmer and Klimchouk, and alsoporosity evolution in mixing zones. The introduction to the geochemical concept has been extended. We furthermore mention and discuss other models of hypogene settings and stress the setting that our model applies to. We refer to earlier modelling efforts that explore this concept in terms of "thermal hypgene speleogenesis" and cross-formational upwelling flow. We have clarified that our work presents a specific hypothetic setting of an unconfined aquifer for both SAS and CAS. We have also pointed out natural settings that our model applies to. An example of Black Hills is also referred in description of the modelling domain. We mention that evolution of porosity in mixing zones has attracted a wider interest in different contexts.

**The model**

We have changed few details in Figure 3 for better clarity.

**Discussion**

To consider the further comments of both reviewers, we have made a deep revision of discussion, which is in mostly rewritten. We again stress the relevant natural settings and discuss the relation to SAS settings. We also point out the limitations of the model and the reasons to keep the settings simple. We compare and discuss this settings to the one presented in 2010 paper (Gabrovšek & Dreybrodt, 2010). To this extend we present and discuss a case when upwelling flow mixes with regional flow under confined settings, which is a response to Comment 6 by Steffen Birk and Comments 3a and 3b by Alexander Klimchouk.

To add some complexity (Comments 3a and 3b by AK) we have added the scenario with embedded prominent fractures.

**Final remarks**

We again thank both reviewers and the editor dr. Neuweiler for all their work and valuable suggestions. We are aware that the model somehow applies for specific settings. However, the outlined mechanisms are surely present in much broader spectrum of realistic settings. We are also aware of the other modelling efforts that demonstrate other mechanisms in hypogenic settings and in mixing zones.

We believe that this manuscript to some extend fills the gap between detailed theoretical studies of processes in the mixing zones and intuitive understanding of speleogenesis.

Although we are aware that some concerns of the reviewers may remain, we hope that the changes and our reasoning fulfill their expectations.

Yours Sincerely,

Franci Gabrovšek & Wolfgang Dreybrodt

---

## Author Response (AR2)

**Dear Editor,**

we are pleased that the manuscript has been accepted. We are aware that it is difficult to find reviewers these days, and we greatly appreciate your editorial contribution. Once again, we thank the two reviewers whose reviews helped improve the paper.

We have done our best to place our work in the context of other work in the revision. We are however concerned that adding more references at this point may lead to excessive referencing.

There is only one correction in the uploaded final version: for consistency, we have changed suplfuric to sulphuric at one point.

Yours Sincerely,
Franci Gabrovšek & Wolfgang Dreybrodt